# Predicting bacterial promoter function and evolution from random sequences

**Mato Lagator[1,2]\*[†], Srdjan Sarikas[2,3][†], Magdalena Steinrueck[2], David Toledo-Aparicio[2], Jonathan P Bollback[4], Calin C Guet[2][‡], Gašper Tkačik[2][‡]**

[1]School of Biological Sciences, Faculty of Biology, Medicine and Health, University of Manchester, Manchester, United Kingdom; [2]Institute of Science and Technology Austria, Klosterneuburg, Austria; [3]Center for Physiology and Pharmacology, Medical University of Vienna, Klosterneuburg, Austria; [4]Institute of Integrative Biology, Functional and Comparative Genomics, University of Liverpool, Liverpool, United Kingdom

**Abstract** Predicting function from sequence is a central problem of biology. Currently, this is possible only locally in a narrow mutational neighborhood around a wildtype sequence rather than globally from any sequence. Using random mutant libraries, we developed a biophysical model that accounts for multiple features of $\sigma^{70}$ binding bacterial promoters to predict constitutive gene expression levels from any sequence. We experimentally and theoretically estimated that 10–20% of random sequences lead to expression and ~80% of non-expressing sequences are one mutation away from a functional promoter. The potential for generating expression from random sequences is so pervasive that selection acts against $\sigma^{70}$-RNA polymerase binding sites even within inter-genic, promoter-containing regions. This pervasiveness of $\sigma^{70}$-binding sites implies that emergence of promoters is not the limiting step in gene regulatory evolution. Ultimately, the inclusion of novel features of promoter function into a mechanistic model enabled not only more accurate predictions of gene expression levels, but also identified that promoters evolve more rapidly than previously thought.

**\*For correspondence:**
mato.lagator@manchester.ac.uk

[†]These authors contributed equally to this work
[‡]These authors also contributed equally to this work

## Editor's evaluation

This paper builds a biophysical model to understand the contribution of the DNA sequence in σ70-RNA polymerase binding activity. The authors provide evidence that taking into account that RNA polymerase can bind in multiple configurations, including non-productive ones, significantly improves predictions. They also confirm and extend previous observations that functional promoter sequences are relatively abundant in random sequences. This work represents an important advance to the field, and will hopefully serve as the stepping stone for better models that include transcription factors and eukaryotic enhancers.

## Introduction

Describing the relationship between sequence (genotype) and function (phenotype) lies at the heart of understanding biology and evolution. Direct experimental characterizations of genotype-phenotype mapping abound (*Lehner, 2013*; *Kemble et al., 2019*), however existing technology limits experimental exploration to only a tiny fraction of all possible sequences (*Sarkisyan et al., 2016*). Due to these technological limitations, there is a need to develop theoretical approaches capable of predicting how any genotype maps onto phenotype (*Yi and Dean, 2019*). Predictive genotype-phenotype maps are rare and incomplete: predicting protein structure from sequence is possible only

for a small number of mutations around a well-characterized wildtype (*Kuhlman and Bradley, 2019*), while predicting RNA folding from sequence (*Schuster, 2006*) generally lacks a relevant reference to the function of that secondary structure.

Gene expression is one of the most fundamental processes of life, and tuning expression levels underpins complex biological function. The critical process in the expression of most bacterial genes is the recruitment of σ70-RNA polymerase (RNAP) to a stretch of DNA – the promoter (*Jacob and Monod, 1961*). Computational and theoretical attempts to predict the relationship between genotype (promoter sequence) and its phenotype (gene expression level) have adopted, broadly speaking, three approaches. (i) Bioinformatics – identifies promoters based on sequence homology to the σ70-RNAP consensus site (*Mustonen and Lässig, 2005*), which consists of –10 and –35 elements (TATAAT and TTGACA in *Escherichia coli*, respectively) separated by a spacer with canonical length of 17 bp, but does not predict gene expression from them (*Anzolini Cassiano and Silva-Rocha, 2020*). (ii) Machine learning – predicts gene expression patterns in cells (*Beer and Tavazoie, 2004*; *Hossain et al., 2020*), but lacks direct links to the underlying biological mechanisms, limiting insights into promoter structure and evolution (*Libbrecht and Noble, 2015*). (iii) Biophysical models – the most successful of which predict gene expression levels based on the thermodynamic properties of σ70-RNAP binding at a promoter in equilibrium (*Bintu et al., 2005*), do not generalize well to random sequences (*Vilar, 2010*). In sum, we lack a generalizable and predictive theoretical and biological understanding of the relationship between promoter genotype and its function (gene expression phenotype) even for constitutive promoters, where σ70-RNAP binding (as opposed to the binding of transcription factors) is considered to be the major determinant of gene expression levels (*Forcier et al., 2018*). Without such a canonical model, we cannot properly understand either promoter function or their evolution.

The standard thermodynamic model (*Bintu et al., 2005*; *Kinney et al., 2010*) considers the energy of binding between σ70-RNAP complex and DNA to identify, typically, the single strongest binding site, with the expression level proportional to the equilibrium occupancy of σ70-RNAP to that binding site ('Standard' model). To calculate equilibrium occupancy, the Standard model assumes that each position in the binding site independently and additively contributes to the binding energy of σ70-RNAP. When constructing the Standard model, the energy contribution of each residue (A, C, G, or T) at each position in the binding site is inferred directly from experimental data and represented in the form of an energy matrix. Constructed in this way, the Standard model represents the state-of-the-art mechanistic model of constitutive gene expression, as well as the traditional view of promoters as containing only a single σ70-RNAP binding site – a view that is at the heart of all bioinformatics approaches to identifying promoters (*Anzolini Cassiano and Silva-Rocha, 2020*).

To predict gene expression levels from any random sequence, we developed a mechanistic and generalizable model based on statistical thermodynamics that expands on the Standard model by accounting for several structural features of bacterial promoters. We use this model, as well as several large promoter mutant libraries, to study the promoter function among random sequences as well as the evolutionary dynamics of bacterial promoters.

## Results

### Extended thermodynamic model

We developed a comprehensive and generalizable thermodynamic model ('Extended model') that expands on the Standard model by accounting for six essential *structural features* of bacterial promoters that are not present in the Standard model (*Figure 1A and B*). (i) The possibility that σ70-RNAP binds the promoter region in multiple configurations that independently and cumulatively contribute to gene expression (*Storz, 2014*; *Belliveau et al., 2018*), as commonly observed in transcriptomics studies (*Srikumar et al., 2015*). We assume that σ70-RNAP concentration is low enough so that the binding of multiple molecules to the promoter has an approximately additive effect on expression. (ii) Spacer length flexibility, assigning an energy penalty for changing the distance between –10 and –35 elements away from the optimum. (iii) Occlusive unproductive binding, occurring when the initiation start site excludes a part of the ribosomal binding site (RBS) resulting in a non-translated transcript. (iv) Occlusive binding on the reverse complement strand, which inhibits productive binding at the promoter (*Brophy and Voigt, 2016*). (v) Dinucleotide interactions between promoter nucleotide

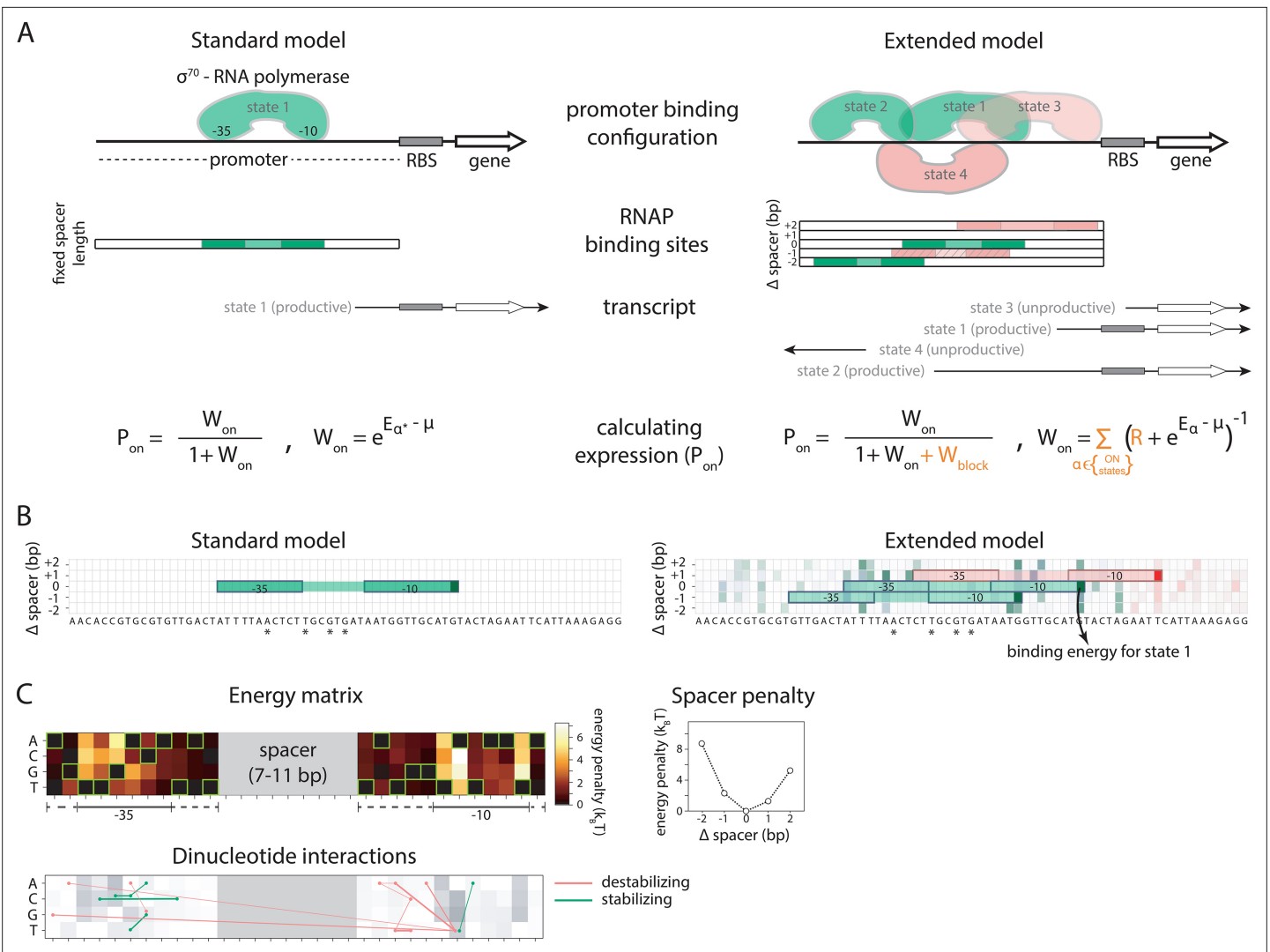

**Figure 1.** Standard and Extended models. (**A**) The standard thermodynamic model assumes only one (strongest) σ⁷⁰-RNAP binding configuration at the promoter, which generates a productive transcript. We refer to 'promoter' as the entire *cis*-regulatory element, while 'binding site' refers to the RNA polymerase (RNAP) contact residues of a specific binding configuration (colored area in 'RNAP binding sites'). The Extended model incorporates structural features of bacterial promoters into the thermodynamic framework: (**i**) cumulative binding, permitting σ⁷⁰-RNAP to bind multiple binding sites independently on the same promoter (binding configurations 1–4); (ii) spacer length flexibility (difference between configurations 1 and 2); (iii) occlusive unproductive binding (configuration 3); (iv) occlusive binding on the reverse complement (RC) (configuration 4). To predict gene expression levels, we calculate the probability of productive σ⁷⁰-RNAP binding ($P_{on}$), where $\mu$ is the chemical potential related to RNAP concentration, $E_\alpha$ the energy of binding for binding state $\alpha$, $R$ the clearance rate relative to σ⁷⁰-RNAP recruitment rate, and $W_{block}$ the binding free energy of unproductive states, which is calculated in the same way as $W_{on}$ but for unproductive as opposed to productive (on) states. The Standard model does not allow for any unproductive binding, and it considers only the single strongest binding site ($\alpha^*$). How the structural features are incorporated into the Extended model is shown in orange. (**B**) Example model output for a selected $P_R$ mutant (stars mark mutated positions) for the Standard and the Extended model, showing binding configuration on the forward strand. Pixels forming the background grid indicate –10 end-points of binding sites, with the intensity of color corresponding to the strength of binding in that configuration (green productive, red unproductive binding). States that are bound strongly enough to independently lead to measurable expression are framed (one for the Standard model, three for the Extended model – two productive, one unproductive). For illustration purposes, the pixel corresponding to the binding energy of State 1 in panel A is marked with an arrow. (**C**) Main biophysical parameters of the Extended model, fitted from *sort-seq* data. Energy matrix shows the effect of every possible binding site residue on the binding energy between σ⁷⁰-RNAP and DNA (strongest binding indicated by green squares). The optimal energy matrix consists of the –10 and –35 elements (underlined), positions outside the canonical elements that significantly affect quantitative predictions of gene expression levels (dotted underline), and spacer of optimal length 9 bp (corresponding to the canonical 17 bp between –10 and –35 elements). Strongest stabilizing (green) and destabilizing (red) interactions between dinucleotides are shown, with line thickness indicating the deviation from independent energy contribution to binding (range 0.15–0.38$k_BT$). For other model parameters and all significant dinucleotide interactions, see *Figure 1—source data 1*. *Figure 1—figure supplement 1* describes the experimental system and protocol; *Figure 1—figure supplement 2* shows the comparison between our and previously

*Figure 1 continued on next page*

*Figure 1 continued*

obtained energy matrix for σ⁷⁰-RNAP.

The online version of this article includes the following source data and figure supplement(s) for figure 1:

**Source data 1.** Inferred values for model parameters.

**Figure supplement 1.** Experimental plasmid systems and protocol.

**Figure supplement 2.** Comparison of σ⁷⁰-RNAP (RNA polymerase) energy matrix values.

residues that are in direct contact with $\sigma^{70}$-RNAP. (vi) Clearance rate of RNAP from the promoter. We do not account for the UP element (*Ross et al., 1993*), as it has an independent role in determining expression (*Einav and Phillips, 2019*) and was beyond the scope of this study. We experimentally verified features (i), (iii), and (iv), as, unlike feature (ii), they have not been characterized before (*Figure 2A–D*). The experimental verification of unproductive binding on the reverse complement was inconclusive – two sets of promoter mutants changed expression in response to increasing strength of binding on the reverse complement, while the other two sets did not (*Figure 2—figure supplement 1*). However, the inclusion of this feature into the Extended model was justified because it led to a significant increase in predictability (especially of the 36N dataset) (*Figure 2D*).

## Experimental mutant libraries and the model

Both, the Standard and the Extended model, were fitted by subsampling a library of >12,000 constitutively expressed random mutants of one of the strongest and best characterized promoters, bacteriophage Lambda $P_R$, that controlled the expression of a *yfp* reporter gene from a small copy number plasmid in *E. coli* (*Figure 1—figure supplement 1A*). In this library, each position in the promoter had a 4% chance of containing each of the non-wildtype bases, resulting in a 12% mutation rate per position. *Sort-seq* experiments were used to measure gene expression levels of mutants (*Kinney and McCandlish, 2019*): mutants were separated using a cell sorter into four phenotypic bins ('no', 'low', 'medium', and 'high' expression), PCR-tagged according to the sorted bin, and bulk sequenced to obtain the counts of each mutant sequence in the sorted bins (*Figure 1—figure supplement 1C-E*). The library was randomly split into three subsets: 'training' (60% of the sequences), used for training model parameters; 'validation' (20%) used for model selection whenever necessary; and 'evaluation' (20%) used exclusively for final evaluation and visualization of model performance.

We inferred model parameters independently for the Standard and the Extended models. For the Standard model, all parameters were inferred under an assumption that only a single (strongest) binding site contributes to expression and that the binding to that site can only contribute positively to expression (i.e. binding of RNAP is never occlusive or unproductive). Note that the Standard model allowed for different spacer lengths but assumed that the energy of binding was not affected by such variations. This already gives our Standard model much more fitting power relative to typical approaches that use a single RNAP energy matrix with a fixed spacer, thereby raising the bar for the Extended model. To infer the parameters for the Extended model, we calculated the binding strengths of $\sigma^{70}$-RNAP to DNA in all possible configurations allowed by the structural features incorporated into the Extended model. This meant, accounting for the energy impact of spacer length variations, the possibility of a promoter containing multiple $\sigma^{70}$-RNAP binding sites, and that the binding to a given site can contribute both positively and negatively to expression. Then, we integrated over all positive ('on') and negative ('off') configurations to obtain the probability ($P_{on}$) of productive $\sigma^{70}$-RNAP binding, which, in this model, solely determines expression levels. For both models, we searched for optimal model parameters (*Figure 1C*) by maximizing the likelihood of the weighted multinomial logistic regression of $\log_{10} P_{on}$ to the median observed expression bin for a training subset (60%) of the $P_R$ mutant library. The energy matrix values obtained by fitting the Extended model correlated strongly with those from a previously published energy matrix of $\sigma^{70}$-RNAP, but also exhibited some systematic differences ($r^2 = 0.72$; $p < 0.001$) (*Figure 1—figure supplement 2*; *Kinney et al., 2010*).

Both models, which were fitted on the training subset of the $P_R$ dataset only, reproduced the evaluation subset of the $P_R$ mutant library with high accuracy, with the Extended model significantly improving on the performance of the Standard model (*Figure 3A and B*). A closer look at the inferred model parameters (*Figure 1—source data 1*) provided novel insights into the mechanisms of RNAP binding, including: (i) the energetic cost of altering the length of the spacer between the –10 and –35

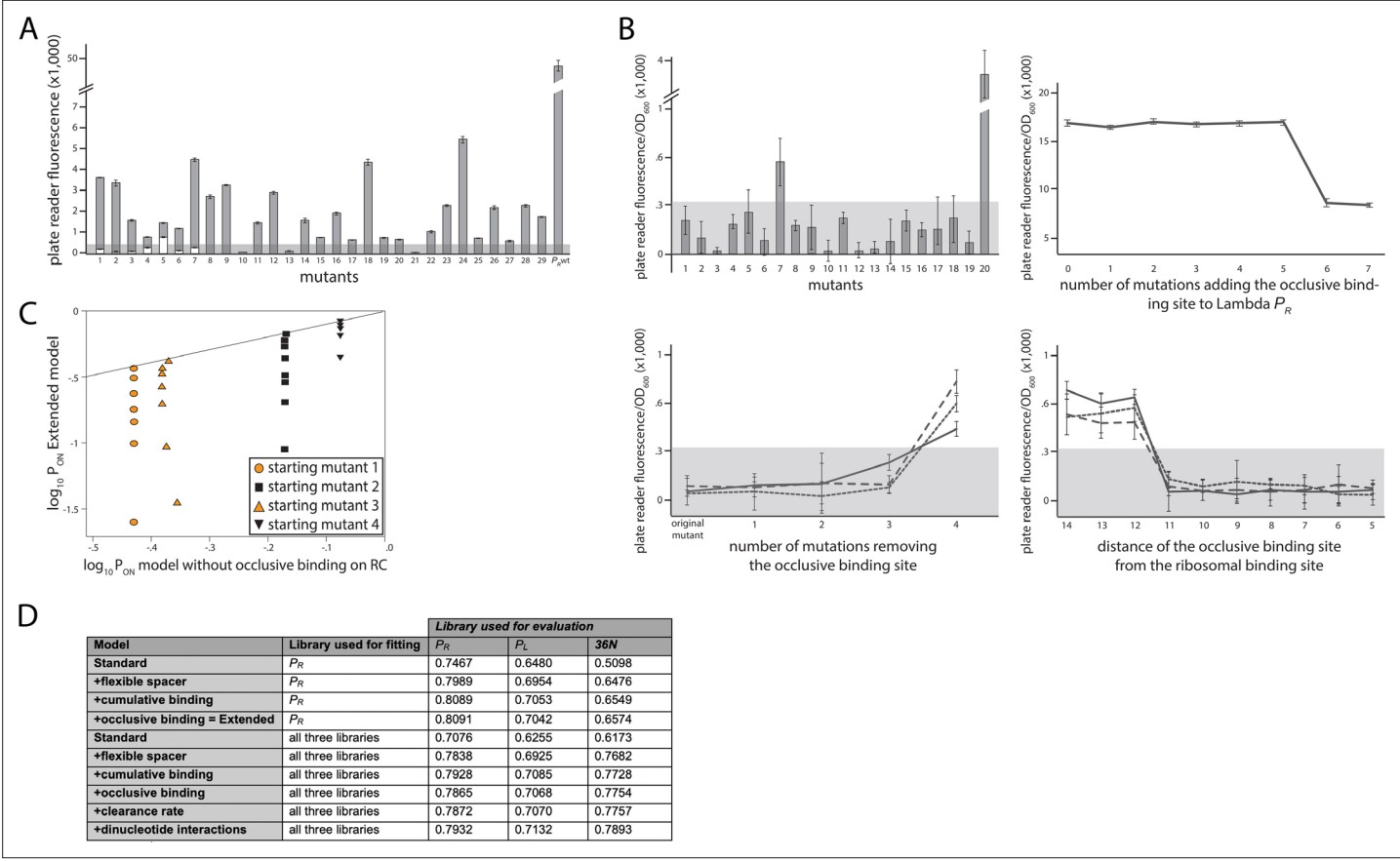

**Figure 2.** Experimental validation of structural promoter features. (**A**) Cumulative binding affects expression in most tested sequences. We experimentally created 29 promoters with the following property: the Standard model predicts no measurable expression from these promoters, while the Extended model predicts measurable expression due to the existence of multiple $\sigma^{70}$-RNAP (RNA polymerase) binding sites. Fluorescence measurements are shown in gray bars, with error bars indicating standard error of the mean from three replicate biological measurements. The gray horizontal bar indicates the detectability ('no measurable expression') threshold, determined for plate reader measurements as the mean fluorescence/$OD_{600}$ value across all replicates of the population carrying the plasmid without any fluorescence markers. All promoters but 10, 13, and 21 exhibited significant measurable expression. We introduced additional mutations into promoters 1–7, in order to remove the secondary binding site(s) without affecting the strongest binding site. White bars show the expression levels of these additional mutants. Only the mutated promoter 5 exhibited significant measurable expression. (**B**) Characterizing sequence determinants of occlusive unproductive binding. (Top left panel) We created 20 promoter sequences for which the Extended model that accounts for occlusive unproductive binding predicted no measurable expression, while the model which did not account for occlusive unproductive binding predicted measurable expression. Bars are mean fluorescence measured from three biological replicates, and error bars are standard error of the mean. The gray shaded area indicates the detectability ('no measurable expression') threshold. Only mutants 7 and 20 exhibited significant measurable expression. (Top right panel) We inserted mutations into the wildtype $P_R$ promoter to gradually introduce an additional binding site that was predicted to bind in an occlusive unproductive manner. These mutations were not predicted to significantly alter $\sigma^{70}$-RNAP binding to the existing dominant $P_R$ binding site. As mutations are introduced into the promoter, they generate stronger binding to the new site, which lowers gene expression levels. (Bottom left panel) We mutated three promoters (originally found in the $P_R$ mutant library) to gradually remove their existing, predicted occlusive unproductive binding sites. As the predicted occlusive unproductive sites were removed, we measured a significant increase in gene expression levels. (Bottom right panel) In order to experimentally verify the occlusive unproductive binding cutoff distance from the –10 end of the binding site to the beginning of the ribosomal binding site (RBS), we started with the same three promoters as in bottom left panel. We used the Extended model to identify the predicted occlusive unproductive binding site, and then we moved the site upstream and downstream to increase or decrease the distance from the RBS, respectively. We identified that a binding site that is 11 or fewer base pairs away from the RBS acts as an unproductive site, while those that are 12 or more base pairs away productively and cumulatively contributed to gene expression levels. (**C**) Mixed support for the role of unproductive binding on the reverse complement in driving expression. We identified four promoter sequences for which we introduced up to eight mutations that would not alter predicted gene expression levels if the model did not account for unproductive binding on the reverse complement, but would if the Extended model was used. In other words, the eight introduced mutations would gradually increase the strength of binding on the reverse complement while having a minimal effect on the strength of binding on the productive strand. Colored points indicate mutants whose measured expression changes in line with Extended model predictions, while black points are mutants whose expression doesn't change compared to the original promoter. Individual responses of each mutant are shown in *Figure 2* – Extended *Figure 1*. (**D**) Improvement to predictability based on each promoter feature. Each structural promoter feature was added to the simpler iteration of the model,

*Figure 2 continued on next page*

*Figure 2 continued*

starting from the Standard model and building progressively toward the Extended. The clearance rate and dinucleotide interactions were included only in the model fitted on all three libraries. The 'unproductive binding' term includes the combined contribution of the occlusive unproductive binding and the unproductive binding on the reverse complement, both of which individually provided a small but significant improvement to model predictions. The values are the fraction of variance explained on the evaluation dataset. *Figure 2 – Extended Figure 2* contains additional verifications of model predictions.

The online version of this article includes the following figure supplement(s) for figure 2:

**Figure supplement 1.** Unproductive binding on the reverse complement.

**Figure supplement 2.** Validation of model predictions.

DNA binding elements, which appears to be approximately quadratic for each deviation away from the optimal 17 bp (spacer penalties in *Figure 1C*); (ii) more precise definition of the binding footprint of RNAP compared to previous RNAP energy matrices (*Kinney et al., 2010*), in particular identifying 11 positions outside the canonical −10 and −35 elements that directly and independently affect $\sigma^{70}$-RNAP (energy matrix in *Figure 1C*); (iii) the role of occlusive unproductive binding of RNAP to the promoter, which reduces overall expression, and whose inclusion into the model significantly improves predictability (*Figure 2D*).

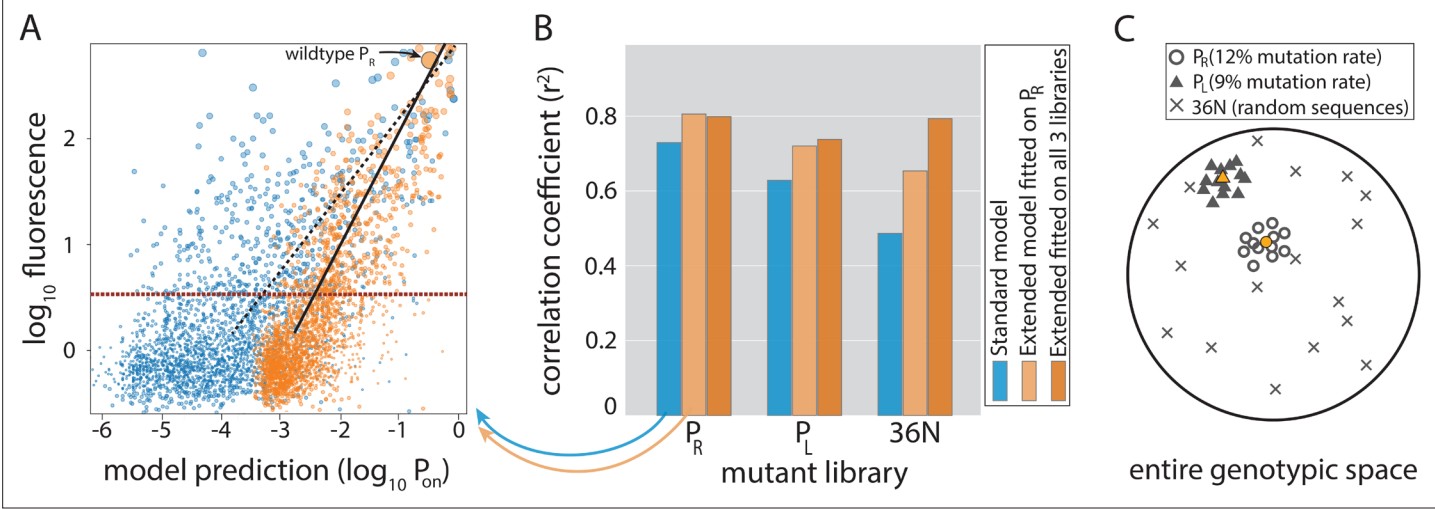

**Figure 3.** Model performance. (**A**) Model-to-data correlation for the Standard (blue) and the Extended model (orange) trained on $P_R$ only, shown for the evaluation subset (20%) of the $P_R$ mutant library. The experimentally measured fluorescence of each mutant is the mean bin (out of four) across all sequenced reads of that mutant (we only consider mutants with at least 30× coverage). Best-fit line (dashed for the Standard model, solid black for Extended) and the instrument detectability threshold, which we refer to as the 'measurable expression' and which marks the 99th percentile of plasmid-free strain (red dashed line), are shown. Marker sizes indicate data-point weights used in fits. We assume that the model predictions are independent of the instrumentally determined measurable expression threshold. (**B**) Model performance on mutant libraries ($P_R$, $P_L$, 36N), shown as fraction of variance explained on evaluation data. Arrows indicate which bars correspond to the correlation plot shown in (**A**). (**C**) Cartoon of the three mutant libraries: $P_R$ and $P_L$ sample locally around a wildtype, with each position having a 12% or 9% chance, respectively, of containing the non-wildtype residue; 36N library contains random 36-bp-long sequences, meaning that it uniformly samples the full 36-bp-long genotypic space (see *Figure 3—figure supplement 2G*). Colored circle and triangle represent the wildtype $P_R$ and $P_L$ sequences, respectively. *Figure 3—source data 1*, *Figure 3—source data 2* provide additional details on the processing of mutant libraries, as does the *Figure 3—figure supplement 2*. *Figure 3—figure supplement 1* shows the performance of the two models on previously published datasets of promoter mutants. *Figure 3—figure supplement 3* shows the plate reader validation of 36N library data processing.

The online version of this article includes the following source data and figure supplement(s) for figure 3:

**Source data 1.** Processing of the mutant libraries and sizes of datasets after splits.

**Source data 2.** Number of mutants per expression bin for each split of the $P_R$, $P_L$, and 36N dataset.

**Figure supplement 1.** Performance of Standard and Extended model on previously published datasets.

**Figure supplement 2.** Processing of mutant libraries.

**Figure supplement 3.** Plate reader validation of 36N data processing.

## Predicting expression from random sequences

To assess the cross-dataset predictability of the two models, we created two additional mutant libraries (*Figure 3C*) with >10,000 mutants each and analyzed them in *sort-seq* experiments. First, we created a library consisting of random mutants of the Lambda $P_L$ promoter, which shares sequence homology to $P_R$. Second, in order to determine model performance across the entire unconstrained genotypic landscape, we built a library consisting of completely random 36 nucleotides controlling the expression of a *gfp* reporter gene ('36N'). In other words, each of the 36 positions had a 25% chance of being either adenine, cytosine, guanine, or thymine. This library was created in a different strain (*Figure 1—figure supplement 1B*), and the expression was sorted into 12 bins instead of 4, giving greater precision in testing the model-to-data fit. Interestingly, ~10% of experimentally measured, random 36-bp-long sequences led to measurable expression (*Figure 1—figure supplement 1F*).

While the Extended model trained on the $P_R$ library only modestly improved predictability of the $P_R$ and local cross-predictability ($P_L$) datasets compared to the Standard model, it dramatically improved predictions of gene expression levels from random sequences (*Figure 3A and B*). This improved performance of the Extended model compared to the traditional view of promoters (the Standard model) indicates that the structural features we accounted for are key components of promoter function. In fact, their inclusion helps overcome one of the major shortcomings of thermodynamic models of gene expression – their low predictability on novel datasets (*Vilar, 2010*), which is observed in Standard model's low predictability of the 36N library.

Each of the structural features included in the Extended model (*Figure 1A*) significantly improved predictability (*Figure 2D*). The contribution of each structural feature to predictability differed between mutant libraries, but, in general, accounting for the energy cost of variable spacer length improved expression level predictions the most. Fitting the Extended model by subsampling all three libraries (>30,000 mutants), instead of just the $P_R$ library, led to further improvements in predictability of expression from random sequences (*Figure 3B*). This resulted in $r^2$ ~0.8, which corresponds to a mean error in expression predictions of 2.8-fold, across a 1000-fold expression range. Fitting the model on all libraries also allowed for a more reliable estimation of dinucleotide interactions between promoter positions that contact $\sigma^{70}$-RNAP (*Figure 1C*). Dinucleotide interactions are moderate contributors to promoter function (*Otwinowski and Nemenman, 2013*; *Figure 2D*), and appear to be stronger when at least one residue is outside of the canonical –10 and –35 binding sites.

The Standard model explicitly assumes that only a single binding position of RNAP governs promoter activity, which we model by identifying the single strongest binding site present in each random sequence in the 36N library (in contrast to the Extended model, which considers all possible $\sigma^{70}$-RNAP binding configurations cumulatively). Because of this assumption of the Standard model, its low cross-predictability (i.e. its performance on fully random sequences – *Figure 3B*) is often attributed to context dependence: the poorly understood effect of positions flanking the –10 and –35 elements (*Vilar, 2010*; *Forcier et al., 2018*). In this study, the ability of RNAP to bind in multiple configurations to the same promoter, rather than a single one, led to the improved predictability of the Extended compared to the Standard model (*Figure 3*). As such, the Extended model explains away the extensive context dependence as a consequence of the promoter structural features that we account for (*Figure 1A*).

## Testing model generalizability

In order to test how well the Extended model predicts gene expression levels from any sequence, we utilized three published, large-scale, promoter mutant libraries whose expression was measured in *E. coli* (*Johns et al., 2018*; *Urtecho et al., 2019*; *Hossain et al., 2020*). Each study also developed a model to predict the variation in gene expression levels observed in their respective datasets, enabling a comparison of performance to our model. As they were generated for different purposes, these libraries contain promoter variants with different features, including but not limited to the features we included in the Extended model (*Figure 1*). We used the Standard and the Extended models with chemical potential as the only physical parameter refitted, meaning that all other model parameters were fitted only from the libraries developed in our study.

*Johns et al., 2018* library consists of ~15,000 promoter sequences harnessed from a wide range of bacterial species and expressed in *E. coli* and contains the most random-like distribution of mutations. As expected, the Extended model ($r^2$ = 0.626) outperformed both the Standard model ($r^2$ = 0.535)

and the model presented in the paper ($r^2 = 0.47$) (*Figure 3—figure supplement 1A*). *Hossain et al., 2020* library contained 4350 sequences with the consensus –10 and –35 elements, as well as a fixed spacer length, with all sequence variability contained within the upstream, downstream, and spacer regions. Both the Extended ($r^2 = 0.602$) and the Standard ($r^2 = 0.598$) models had high performance on the dataset compared to the model presented in the paper ($r^2 = 0.45$), indicating that while the conserved consensus –10 and –35 elements drove a bulk of the observed expression variation, we nevertheless extracted significant predictability gains due to the extended RNAP binding footprint (*Figure 3—figure supplement 1B*). *Urtecho et al., 2019*, developed ~11,000 promoters consisting of all combinations of a set of modules: eight –35 elements, eight –10 elements, three UP elements, eight spacers, and eight backgrounds. The Extended ($r^2 = 0.599$) slightly outperformed the Standard model ($r^2 = 0.588$), suggesting that the eight background sequences contained different secondary $\sigma^{70}$-RNAP binding sites that cumulatively contributed to expression (*Figure 3—figure supplement 1C*). However, both models underperformed the machine learning approach presented in the paper ($r^2 = 0.955$) as the expression variation from these modular promoters was driven by interactions between components (*Urtecho et al., 2019*), which were not captured by the Extended model. It is important to emphasize, however, that the machine learning model developed by Urtecho et al. considers only combinations of a small number of predefined sequence elements, and as such cannot predict expression level from arbitrary sequences.

Together, these analyses point to two important aspects of promoter function. First, the high performance of the Extended model on most tested datasets without refitting (comparable to the performance on the 36N library of the Extended model trained only on the $P_R$ library), combined with its high predictability of expression from random sequences (the 36N library), suggests that the binding energies and the structural promoter features described in this study (*Figure 1*) are important quantitative determinants of promoter function. Second, while these features are sufficient to predict expression levels from many sequences, especially if sequences are random, the Extended model failed to capture the variation arising from the interactions between various promoter features (*Urtecho et al., 2019*). The mechanisms that govern the way in which different promoter features – beyond those captured by the Extended model – interact with each other remain largely unclear – but see *Einav and Phillips, 2019*.

## Constitutive promoters evolve readily from random sequences

Having a biophysical model capable of predicting gene expression levels from random sequences enabled us to computationally sample the entire genotypic space in order to describe the genotype-phenotype mapping of constitutive promoters. Approximately 20% of random 115-bp-long sequences (average inter-genic region length in *E. coli*) are predicted to have measurable expression, compared to ~8% for the Standard model (*Figure 4A*). Surprisingly, for ~82% of non-expressing random sequences, at least one point mutation could be found that would led to measurable expression (*Figure 4B*) – a finding that we verified experimentally (*Figure 2—figure supplement 2A*). In total, more than 1.5% of all possible point mutations converted a random non-expressing sequence into a functional promoter (*Figure 4C*), indicating that promoter sequences readily emerge, as previously suggested based on a limited sample of ~40 sequences (*Yona et al., 2018*).

To contextualize these findings, we simulated evolution under directional selection for expression from random non-expressing sequences using a Strong-Selection-Weak-Mutation model (*Gillespie, 1983*) adapted from *Tuğrul et al., 2015*. In short, this model considers one mutation at a time: as one mutation arises randomly in the sequence, the model determines the probability of that mutation getting fixed in the population or eliminated due to selection and drift, and only then is the fate of the next random mutation considered. This evolutionary scenario probes the upper speed limit for the emergence of new functional promoters. The Extended model predicts, on average, more rapid evolution from a non-expressing sequence (*Figure 4—figure supplement 1A*), toward either weakest measurable expression or strong (wildtype $P_R$ level) expression, when compared to the Standard model (*Figure 4D*, *Figure 4—figure supplement 1B*). The difference in rates of evolution becomes even more pronounced as the mutating sequence gets shorter (simulating stronger constraints due to, for example, existence of other transcription factor binding sites) (*Figure 4D*). The increased rate of promoter evolution predicted by the Extended, as compared to the Standard model, means that promoters can evolve even more rapidly than expected based on the standard RNAP consensus

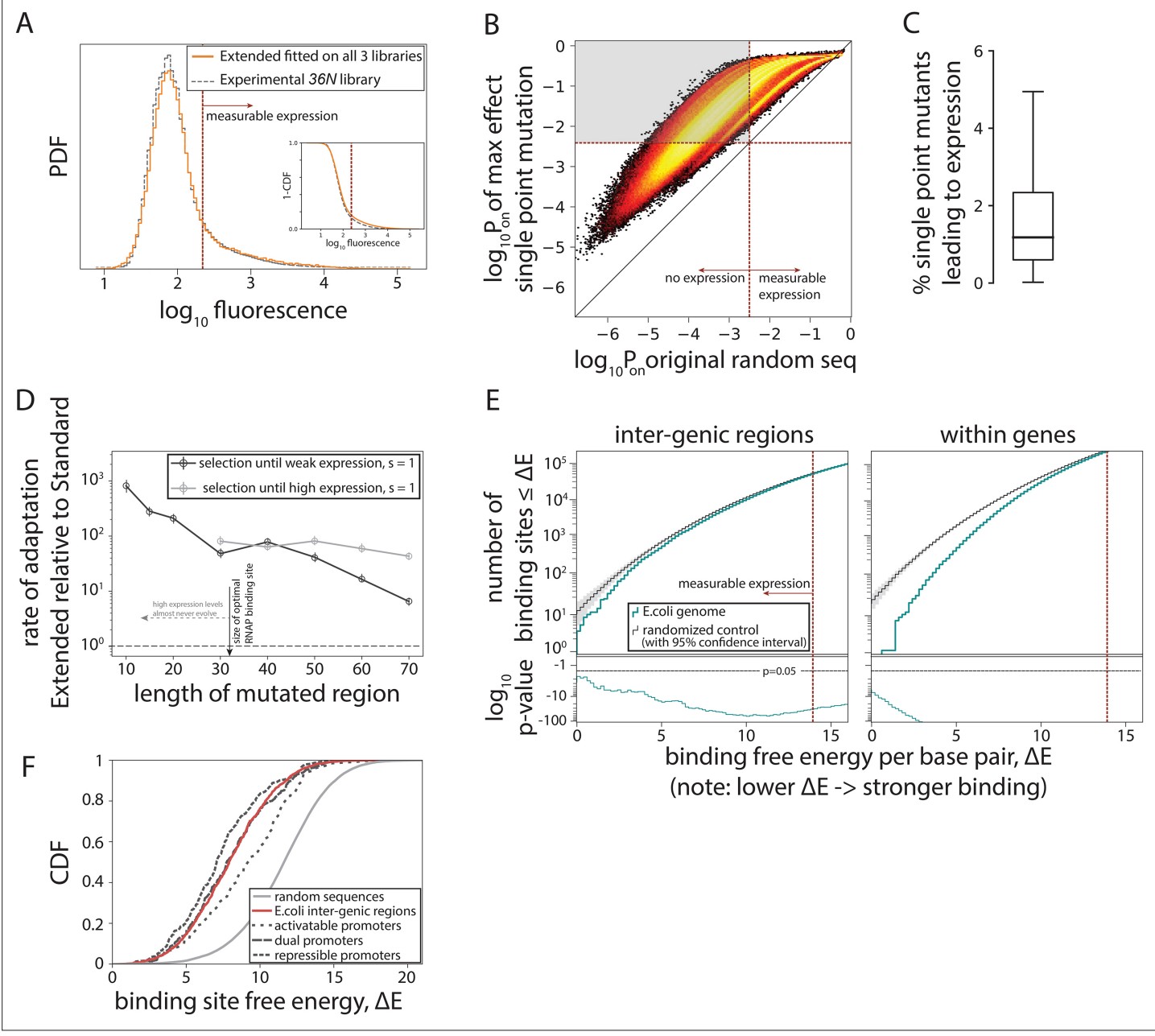

**Figure 4.** Evolution of promoters. (**A**) Probability density function (PDF) for the flow cytometry measurement of the 36N library (gray dashed line) compared to the flow cytometry fluorescence intensities simulated from $10^6$ randomly generated 115-bp-long sequences using the Extended model fitted on all three libraries. Red dotted line marks the cutoff for 'measurable expression', estimated from experimental data. Measurable expression is defined to correspond to the 99th percentile of fluorescence measurements of the experimental strain carrying no plasmid and, hence, no fluorescence. Inset: cumulative distribution function (CDF) for the same comparison. (**B**) Density heat map (brighter color represents higher density), showing, for every simulated random sequence (expression on x-axis), the expression of a single point mutant with the largest positive effect on predicted expression ($P_{on}$) (y-axis). For 82% of non-expressing random sequences (sequences left from the dotted line on the x-axis), that mutation led to measurable expression (gray area). (**C**) Box plot showing the percentage of all possible point mutations predicted to convert a given random non-expressing sequence into one with measurable expression (obtained from $10^5$ random sequences). (**D**) Increase in rates of adaptive evolution of the Extended relative to the Standard model. Evolution to either weakest measurable expression, or high ($P_R$) expression levels was modeled through single point mutations. Evolution was simulated 100 independent times for each of the 100 random 115-bp-long starting sequences, by mutating the central contiguous part of the indicated length. Evolving promoters would almost never reach high expression levels when only a region smaller than the RNA polymerase (RNAP) binding site (30 bp) was allowed to mutate. (**E**) For evidence of selection against σ70-RNAP binding sites, we compared the free energy per bp between either the inter-genic (typically, promoter containing) or the within-genic regions of the *Escherichia coli* genome, and that of a random sequence with the GC% of the corresponding region (note that higher energy means weaker binding and hence lower expression). At lower binding energies (corresponding to

*Figure 4 continued on next page*

*Figure 4 continued*

stronger binding), the actual number of binding sites in the *E. coli* genome (teal) is lower than expected based on random sequences (gray). Associated p-values are also shown. The total number of binding sites increases with binding energy (i.e. there are a lot more weaker than stronger binding sites), explaining the variability in p-values. (**F**) CDFs for predicted binding strengths of different *E. coli* promoters, obtained from RegulonDB. *Figure 4—figure supplement 1* shows further details on how promoter evolution was modeled. *Figure 4—figure supplement 2* contains the information about the contribution of cumulative binding to expression. *Figure 4—figure supplement 3* shows additional tests for selection against σ⁷⁰-RNAP binding sites.

The online version of this article includes the following figure supplement(s) for figure 4:

**Figure supplement 1.** Modeling evolution.

**Figure supplement 2.** Cumulative binding contributes more to expression at weak promoters.

**Figure supplement 3.** Further tests of selection against σ⁷⁰-RNAP (RNA polymerase) binding sites.

sequence or the corresponding energy matrix. Under certain conditions, the rate of increase is several orders of magnitude. This increase arises from the inclusion of the new structural promoter features (*Figure 1A*), which dramatically enlarges the number of possible productive binding configurations – a phenomenon particularly evident at lower expression levels, where cumulative binding across multiple binding sites substantially contributes to overall expression (*Figure 4—figure supplement 2*).

## Selection against RNAP binding sites in *E. coli* promoters

The pervasiveness of σ⁷⁰-RNAP binding sites among random DNA sequences has surprising implications for understanding the evolution of gene regulation: the critical question is no longer how do promoters arise, but rather how does a cell avoid expressing everything, all the time. To address this, we asked if there was evidence of selection against σ⁷⁰-RNAP binding sites in the *E. coli* genome (*Figure 4E*, *Figure 4—figure supplement 3A*; *Mustonen et al., 2008*). While identifying strong selection against σ⁷⁰-RNAP binding sites within genes (*Yona et al., 2018*), surprisingly, we also found evidence of selection against binding sites in inter-genic regions (p < 10⁻⁴⁰ across all inter-genic regions). This negative selection was observed even when only the known, experimentally determined σ⁷⁰ promoters (obtained from RegulonDB; *Salgado et al., 2013*) were considered (*Figure 4—figure supplement 3B*), most of which strongly bind σ⁷⁰-RNAP and hence lead to intermediate and high expression levels (*Figure 4F*). Thus, there is selection against σ⁷⁰-RNAP binding sites not only within genes, but also in the inter-genic regions that contain active promoters. This selection against too many binding sites in promoters could have arisen from a direct competition between σ⁷⁰-RNAP and other transcription factors, and/or in response to the potential, and poorly understood, cost for a promoter to contain multiple weaker (as opposed to fewer stronger) σ⁷⁰-RNAP binding sites.

## Discussion

While DNA sequence ultimately encodes all the genetic elements for the function of living systems, decoding this function directly from sequence remains a grand challenge. Despite limited success, for example, in connecting the effects of mutations on protein or secondary RNA structures (*Schuster, 2006*; *Dill and MacCallum, 2012*), we lack a quantitative link between most cellular processes and DNA sequence.

One of the most fundamental cellular processes is the regulation of gene expression, which is governed by the binding of *σ* factors, RNAP, and transcription factors to promoters. Yet, we lack a way to decode promoters: a code or dictionary that would map promoter sequences into their function – the level of gene expression (*Kinkhabwala and Guet, 2008*). This mapping is fundamental for understanding gene regulatory network structure and their evolution (*Voordeckers et al., 2015*; *Lässig et al., 2017*), and for engineering synthetic biological systems (*Kim et al., 2009*). Moreover, gene expression is a phenotype that captures our intuitive notion of promoter function, enabling simulation of evolutionary scenarios that select for that function – a feat that appears unrealistic for cases where a phenotype has no obvious functional significance. Accounting for structural features of promoters (*Figure 1A*) improved the predictions of gene expression levels from any DNA sequence, by extending the applicability of thermodynamic modeling toward low expression levels and beyond local neighborhoods of known promoters (*Vilar, 2010*; *Forcier et al., 2018*), to capture the entire genotypic space. This is a necessary precondition to ask questions about de novo promoter evolution.

The Extended model also allowed us to better understand key mechanisms of promoter function, such as the existence of multiple binding sites that cumulatively contribute to expression (*Figure 4—figure supplement 2*). Recently, a complementary approach described two further structural features of DNA-RNAP interaction, the avidity between –10 and –35 binding sites and the UP element (*Einav and Phillips, 2019*), yet this approach was not designed to predict expression from random sequences. Integrating these and other additional mechanisms of RNAP and promoter function (*Hawley and McClure, 1983*; *Roy et al., 1998*) into our Extended model could further improve promoter sequence-to-function predictions. Furthermore, the thermodynamic framework can be extended to model complex, regulated promoters that bind transcription factors, as long as their biophysical parameters (energy matrices) are known (*Bintu et al., 2005*; *Saiz and Vilar, 2008*).

The flexibility of $\sigma^{70}$-RNAP binding ensures the proximity of any DNA sequence to a constitutive promoter (*Figure 3*). This permits easier evolutionary tuning of gene expression levels, while also allowing $\sigma^{70}$-RNAP to function as a de facto global regulator (*Salgado et al., 2013*; *Igler et al., 2018*). Selection can thus sustain the binding specificity of required transcription factors and reduce crosstalk while maintaining expression (*Friedlander et al., 2016*). The ease of evolving constitutive promoters, which has been experimentally hinted at using a very limited number of mutants (*Wolf et al., 2015*; *Yona et al., 2018*), remained theoretically puzzling (*Tuğrul et al., 2015*). In fact, Tugrul et al. failed to identify mechanistic or evolutionary factors that would increase the predicted speed of binding site evolution and bring it closer to experimental observations. To address this discrepancy, we linked an extended biophysical model of promoter function, inferred directly from large mutant libraries, with a quantitative evolutionary framework to show that promoters indeed evolve more rapidly than expected based on existing models of protein-DNA interactions. Strikingly, this increase in the rates of promoter evolution, which can be orders of magnitude faster in the Extended compared to the Standard model, arises from the same promoter features that account for only a 2-to-3-fold increase in the likelihood of random sequences resulting in measurable expression. We hypothesize that this potentiation comes about due to a vast expansion in the number of accessible evolutionary paths that lead from a random initial sequence to the functional promoter, likely created by the flexible spacer, which is an interesting question for future research. Another key question that arises from our results is how $\sigma^{70}$-RNAP navigates the fine balance between evolvability of binding sites and their specificity, the latter hinging on selection against spurious binding. Our study shows how biophysical models that accurately capture mechanisms of biological function offer a robust method for addressing key questions in biology and evolution.

# Materials and methods

## Experimental systems and mutant libraries

The $P_R$ system consisted of *venus-yfp* (*Nagai et al., 2002*) under the control of Lambda bacteriophage $P_R$ promoter (*Figure 1—figure supplement 1A*). The system was isolated from the rest of the plasmid with two strong terminators, T1 and T17, obtained from iGem Parts registry. The $O_{R3}$ site of $P_R$ was removed in order to remove the $P_{RM}$ promoter. The RBS (carrying the AGGAGG Shine-Delgarno sequence) was 28 bp away from the downstream end of the strongest $\sigma^{70}$-RNAP binding site in the $P_R$ promoter. The entire cassette was inserted into a low-copy number plasmid backbone SC101* carrying a kanamycin resistance gene (*Lutz and Bujard, 1997*). The random $P_R$ mutant library was created by cloning custom-made oligonucleotides (IDT Technologies), which had a 12% mutation rate for each of the 67 positions in the $P_R$ promoter (4% mutation chance for each possible mutation away from the wildtype). The mutagenized region concluded 8 bp upstream of the wildtype $P_R$ promoter transcriptional start site, and hence that start site was not mutated. Ligated plasmids were electroporated into One-Shot Top10 electrocompetent *E. coli* cells (Life Technologies, Carlsbad, CA). This step was used to maximize the library diversity due to One-Shot Top10 cells' high competency. Following electroporation, cells were grown for 1 hr in LB broth and plated on selective LB plates with 50 μg/mL kanamycin to allow single colony formation and minimize resource competition, and were then grown overnight. To ensure large coverage, we cloned mutagenized PCR products until we obtained at least 30,000 individual colonies (uniquely transformed individuals). We obtained this number of individual colonies by estimating the number of colonies on each selective plate, and then repeating the above cloning procedure until each library contained at least 30,000 individual colonies. Using

chilled LB media, colonies were washed off plates and collected. Plasmids were isolated from this collection in bulk using a Qiagen Mediprep kit, and transformed into strain K12-MG1655. The same wildtype layout and mutagenesis protocol was used to create the $P_L$ mutant library, with the exception that mutation rates were 9% per nucleotide (with 3% mutation chance for every possible non-wildtype mutation). Cells were always grown in a shaking incubator at 37°C.

The 36N library was placed in pUA66-lacZ plasmid backbone carrying Kan resistance (*Zaslaver et al., 2006*), by cloning a 100-bp-long oligonucleotide containing 36 randomized base pairs (each of the 36 positions had a 25% chance of being either adenine, thymine, cytosine, or guanine), surrounded on each side by 32 bp of randomly generated, unmutated, non-expressing DNA sequence (*Figure 1—figure supplement 1B*). The two 32-bp-long segments had a different sequence, as each was randomly generated. This 100 bp sequence was placed upstream of the RBS, controlling the expression of a *gfp* gene (*Zaslaver et al., 2006*). The rest of the plasmid had the same components as the one used for the $P_R$ library. The 36N plasmid DNA library was generated using a Q5 site directed mutagenesis kit (New England Biolabs, Ipswitch, MA). For amplification, we used the reference plasmid as a template (a plasmid with a random, non-expressing 100 bp fragment) and two pools of primers with a constant 3' end and an 18N random 5' end. We cloned the mutagenesis products into electro-competent NEB5α cells. Following electroporation, cells were grown for 1 hr in LB and then plated overnight on selective kanamycin plates. Cells were collected in bulk to form the 36N mutant library. We used different strains between the $P_R$ and the 36N libraries in order to better evaluate and increase the generalizability of our model. The mutation rates of all three libraries (*Figure 1—figure supplement 1G*), as well as the sequence-level randomness of the 36N library, were checked based on the *sort-seq* data. The libraries also contained promoter sequences with a range of spacer lengths (*Figure 1—figure supplement 1H*).

## *Sort-seq* Experiments

Prior to sorting, cells were grown in M9 minimal medium with 0.2% (w/v) Casamino acids, 0.2% (w/v) glucose and 50 µg/mL kanamycin. Frozen aliquots of each mutant library were diluted 1:100 and grown overnight. Prior to sorting, overnight cultures were diluted again 1:100 and grown until exponential phase (for 3 hr in M9 media). We repeated the sorting of each library in three biological replicates.

Fluorescence activated cell sorting (FACS) was performed on a FACS Aria III flow cytometer (BD Biosciences, San Jose, CA) with a 70 µm nozzle. A 488 nm laser was used to detect forward scatter (FSC) and side scatter (SSC) with a 488/10 band-pass filter. FITC channel was used for excitation of either YFP ($P_R$ and $P_L$ libraries) or GFP (36N library). The flow rate was set to 1.0 and samples were diluted to obtain a cell count of approximately 2000 events/s. Cells for sorting were manually gated on the densest population in an FSC/SSC scatter plot, which comprised 95.5% of all events exceeding a threshold of 1000 on the SSC axis. For the $P_R$ and $P_L$ libraries, four sorting gates were set on FITC: no-expression gate, capturing >99% of all measurements from a non-expressing plasmid (control plasmid not containing *yfp*); high-expression gate, capturing >99% of all measurements from the wildtype $P_R$ plasmid; two gates equidistant in fluorescence between the no- and high-expression gates (*Figure 1—figure supplement 1F*). The boundary for the no-expression gate (obtained from the SC101* strain without the fluorescence marker) was used to define the 'measurable expression' threshold for $P_R$ and $P_L$ libraries. The 36N library was sorted into 12 gates as follows: The upper boundary of the lowest gate corresponded to the median of an auto-fluorescence control sample (plasmid-free Top10 cells). The upper boundary of the second-lowest gate, which also defined the 'measurable expression' threshold for the 36N library, was the 99th percentile of the fluorescence-free cells. The lower boundary of the highest gate (B12) was set to $2 \times 10^4$. Distances between the remaining intermediate nine gate boundaries were of equal size on the log-scale FITC histogram (*Figure 1—figure supplement 1F*). For $P_R$ and $P_L$ libraries we could sort into all four bins simultaneously, and hence we sorted 1 million cells for each of the three biological replicates. For the 36N library, we first recorded $10^5$ reads, and then the number of cells sorted into each of the 12 bins corresponded to the number of cells recorded in each of the bins. The recipient plate was cooled to 4°C to halt growth while sorting to other wells was still going on. Only for the 36N library, after sorting we added 1000 cells with the reference plasmid (experimental $P_R$ plasmid) into each of the 12 sorted populations. We did this to maximize the precision of our experimental measurements for the 36N library, as it would enable more accurate experimental determination of gene expression levels by

enabling normalizing the number of mutants in each bin (see section *Processing of the 36N mutant library*).

Cells from each sorted bin were grown overnight. We isolated plasmids from the sorted populations. We used high-fidelity PCR (Phusion, New England Biolabs) to amplify 150 bp containing the mutagenized region, and barcoded the primers according to the sorted bin. Four sets of barcoded primers were used for the $P_R$ and $P_L$ libraries, and 12 for the 36N library. PCR products were column-purified (Zymo Research, Irvine, CA) and eluted in 30 μL, of which 2 μL were run on an agarose gel for product quantification based on band fluorescence. PCR products were pooled to reach approximately equimolar concentrations of each bin, separately for each mutant library. No additional clonal amplification steps were conducted prior to sequencing. Each library was sequenced with millions of reads using 135 bp pair end Illumina sequencing (Hi-seq).

The nature of the 36N library – it containing completely random 36-bp-long sequences – means that we might not be measuring the effects of each random sequence on $\sigma^{70}$-RNAP binding, but also possibly on other factors that impact gene expression levels, most important of which could be mRNA stability. Our experimental setup does not allow us to directly disentangle the effects of each individual sequence on mRNA stability versus binding energy. On the other hand, the $P_R$ and $P_L$ libraries have a clearly defined wildtype sequence around which all mutations were introduced. Introducing a relatively small number of mutations into these promoters is unlikely to dramatically alter mRNA stability (*Mohanty and Kushner, 2016*). The fact that the Extended model trained only on the $P_R$ dataset performs so well on the 36N library (*Figure 3B*) makes us confident that the majority of 36N mutants predominantly affect $\sigma^{70}$-RNAP rather than mRNA stability. In other words, while the primary mechanism linking sequence to gene expression might be mRNA stability for some mutants in the 36N library, on average the effect of mutants on binding energy dominates.

## Processing of $P_R$ and $P_L$ mutant libraries

Here, we describe the data processing pipeline for the $P_R$ and $P_L$ libraries, from the initial sequence reads to a dataset we use for fitting and evaluating our model. For each library, we obtained millions of reads (*Figure 3—source data 1*), which we paired (using the illumina-utils package; *Eren et al., 2013*), discarding reads with any mismatch. Each read contained a tag on both ends with information of the expression bin the sequence was sorted into, and we only included reads which have the same tag on both ends.

In each library, the remaining ~2.5 million reads contained more than 300,000 unique sequences, which we filtered based on length, coverage, and the position of the RBS (Shine-Dalgarno sequence that was the same for all sequences in the library as it was not mutated) (*Figure 3—figure supplement 2A*,B). We further required the sequences not to be too different from the ancestral sequence ($P_R$ and $P_L$, respectively) (*Figure 3—figure supplement 2C*). This step removed the remaining $P_R$ sequence and a small cloud of sequencing errors around it from the $P_L$ library. $P_R$ wildtype sequences were present in the $P_L$ library because both libraries were made using the $P_R$ wildtype sequence as the starting point for cloning and ligation – a technique that is nearly but not exactly 100% efficient. Even though we used high-fidelity polymerase for the PCR, still some sequencing errors existed (as evident from the 'cloud' of sequencing errors around the $P_R$ wildtype in the $P_L$ library), but were lower than 0.01%. Finally, we required the distribution of expression bins to be as unambiguous as possible (*Figure 3—figure supplement 2D*,E). This requirement was introduced because *sort-seq* experiments involve several passages of bacterial cultures after they have been sorted into their corresponding bins. This passaging can introduce a bias in the form of over- or under-representation of a given mutant in one bin compared to its abundance in the adjacent bins. Approximately 0.5% of all sequences had highly unequal distributions between adjacent bins, and were thus excluded.

We observed a discrepancy in the number of unique sequences in the two libraries (*Figure 3—source data 1*) that arose due to the higher mutation rate of the $P_R$ library. This is why we took a more conservative approach and additionally raised the coverage threshold to 30 for all our analyses.

## Processing of the 36N mutant library

The original sequences library contained more than 10 million paired reads, almost all of which paired without mismatches (*Figure 3—source data 1*). Out of those, more than a million contained the reference plasmid sequence, which we extract and treat separately. We required sufficient mapping

(higher than 0.75 similarity, using local alignment function from the pairwise2 module in Biopython, with symmetric gap open and extension penalty of $-(2L)^{-1}$, and a matching score of $L^{-1}$, where $L$ is the length of the mapping region) of both flanking regions, and that the length of the core region is within 2 bp of the canonical (36 bp) one. Most of those reads were unique and most probably errors, so we originally required coverage of at least two (*Figure 3—source data 1*).

At this point we set to estimate the abundance of spurious sequences (e.g. sequencing or PCR errors), by cross-mapping the abundant sequences (with the highest coverage) against all other sequences, using the same scores as defined above (*Figure 3—figure supplement 2F*). In every case the distribution of mapping scores was bimodal, with an overwhelming number of low scores and a small number of very high scores, suggesting that abundant sequences come with a cloud of errors, which are similar to the investigated sequence. Compiling a distribution of similarity scores for the 1000 most covered sequences showed a clear threshold of 0.7 between the low- and high-scored modes. We then accumulated all the sequences that appeared to be associated with a number of high-covered sequences and noted that the histogram of coverages of only those sequences closely matches the low-end part of the histogram of all sequences. Extrapolating from this, we estimated that requiring a coverage of at least 30 reads would lower the number of spurious sequences to only a few dozen (*Figure 3—figure supplement 2G*).

In contrast to the $P_R$ and $P_L$ libraries, where FACS was performed simultaneously into four bins, here the experiment involved sequential sorting of the same number of cells into one of the 12 bins (*Figure 1—figure supplement 1F*). Naturally, less abundant expression bins had to be sorted for a longer time, which introduced a systematic bias for higher expressing bins. To account for this bias toward higher-expressing sequences, we introduced during the cell sorting step to each bin a known number (1000) of cells carrying the reference $P_R$ plasmid. Note that this was not the same plasmid as the one used for cloning the 36N library, allowing us to easily distinguish between the reference plasmid and the wildtype plasmid that was not cloned successfully (i.e. was not carrying any mutations).

To debias, we divided each distribution by the distribution for the reference sequence (*Figure 3—figure supplement 2H*), and normalized. We then constructed a set of average distributions for all sequences that have the same mode. This allowed us to fit the background noise (intrinsic to FACS measurements) for all distributions. Next, we cleaned the sequence-specific distribution of potential outliers that may distort the estimate of the mean: for each mode-specific template distribution, we defined a filter that selected only bins in which the background is at max ⅓ of the value of the template. For each sequence-specific distribution, we selected a filter based on its mode, which nullified values in bins defined as outliers (*Figure 3—figure supplement 2I*,J). Afterward, we renormalized the distribution. This filtering is especially important for the higher-expressing sequences, where a single count in the lower bins would get enlarged tremendously by debiasing, drastically skewing the expression estimate (*Figure 3—source data 1*).

We used the debiased and filtered distribution over bins $\alpha_i = [\alpha_0, \ldots, \alpha_{11}]$, to produce two estimates of gene expression. In the units of bin index, we estimate expression as $\varepsilon_{\mathrm{bin}} = \sum_{i=0}^{i \leq 11} a_i \cdot i$, while in the units of luminosity measured in FACS as $\varepsilon_{\mathrm{facs}} = \sum_{i=0}^{i \leq 11} a_i \log_{10} m_i$, where $m_i$ is the median value of measurements in the $i$-th bin. The two estimates are linearly related.

To validate our data processing, we randomly picked eight mutants from each of the 12 bins, sequenced them and measured in a plate reader. Out of 96 mutants, 79 were unique, and 2 did not exist in the 36N *sort-seq* library. We compared the measurements of the remaining 77 sequences obtained from a plate reader to the estimate of their expression obtained from the *sort-seq* experiment following the above-described debiasing and filtering (*Figure 3—figure supplement 3*). Plate reader measurements were conducted in the following manner: mutants were grown overnight in M9 minimal media supplemented with 0.2% (w/v) CAS, 0.2% (w/v) glucose, and 50 μg/mL kanamycin. The overnight populations were diluted 1000-fold, grown until $OD_{600}$ of approximately 0.1, and their fluorescence measured in a Bio-Tek Synergy H1 plate reader. Three replicates of each mutant were measured.

## Data splits and fitting procedure

We define 60:20:20 percentage splits of each of our libraries into three disjunct datasets (*Figure 3—source data 2*). The first ('Training' dataset) and second ('Validation' dataset) were used for training and model selection respectively, while the last ('Evaluation' dataset) was used exclusively for final

evaluation and visualization of our models. Sequences were randomly divided into three splits. Note that each sequence in our library is unique, thus no sequences were repeated between libraries.

The central quantity of our modeling approach is the proportion of time RNAP spends bound to the DNA in the 'on' configuration, $P_{on}$, that is, in a configuration that can yield productive mRNA and thus lead to protein expression. Our quantification of expression, which we define as log fluorescence, will then be proportional to $\log_{10} P_{on}$ (**Bintu et al., 2005**; **Kinney et al., 2010**).

The three libraries were different in terms of their output. While for the $P_R$ and $P_L$ libraries we used only four bins and chose the median bin for each mutant (more conservative statistic), we obtained more plausible estimates of expression for the mutants in the 36N library (as explained above). To keep the procedure the same across all three libraries, for the 36N library we rounded off the estimate of the mean expression in bin units to the closest integer. This allowed us to use multinomial logistic regression for all three datasets and the associated log-likelihood as the objective function.

Concretely, given a set of parameters (binding matrix, spatial penalties, chemical potential, potential dinucleotide interactions), our model produces $\log_{10} P_{on}$ for each sequence, which we use as an independent variable to fit observed bin expression levels using multinomial logistic regression (from scikit learn, with L-BFGS-B optimization; **Byrd et al., 1995**). When implementing multinomial logistic regression, we treat $\log_{10} P_{on}$ as an independent variable ($x = \log_{10} P_{on}$), so that the log-odd of a sequence falling into a bin is a linear function of $x$: $\log \frac{\pi_i}{1-\pi_i} = a_i x + b_i$ . This yields individual probabilities: $\pi_i = \frac{1}{1+e^{-(a_i x + b_i)}}$. In the case of multi-label classification, $P_i$ needs to be further normalized by 1, so that the probability associated with $x$ being measured in bin is: $P_i = \frac{\pi_i}{\sum_i \pi_i}$ . Additionally, we explicitly required a balanced fit by applying weights inversely proportional to the number of observations in each bin for each dataset. This is especially important for the 36N library, due to the highly disproportionate numbers of observations in the 12 bins.

The main metric for our optimization and model performance is the likelihood of the logistic regression. In the interest of higher interpretability, we also report the $r^2$ value of a linear fit where the independent variable is $\log_{10} P_{on}$ and the dependent variable is the log fluorescence estimate, using the same weights as for the logistic regression. The fluorescence estimate is the median for the $P_R$ and $P_L$ libraries, as a robust measure in the absence of high bin resolution; and mean for the 36N library, as having 12 bins allowed us to accurately estimate the mean. The multinomial logistic regression does not necessarily yield a linear dependence between $\log_{10} P_{on}$ and observations. In fact, across the range of reasonable values, the log-likelihood depends on the chemical potential only weakly, yet it may change the correlation coefficient by several percent. For this reason, in **Figure 3B**, we show correlation coefficients ($r^2$) for evaluation datasets with chemical potentials re-optimized on the respective training datasets. Raw $r^2$ values are similar and are reported in **Figure 2D**.

We start from the Standard model, and progress toward the Extended model sequentially, including one structural feature at a time. We fit the model parameters using only the training dataset of the $P_R$ library, and only search in the vicinity of the previous best fit. We assess such a model on the $P_R$ evaluation and all three datasets of the $P_L$ library without any adjustments, since the two libraries were obtained following the same experimental protocol in the same cells. In that sense, the whole $P_L$ library played the role of an evaluation set. To evaluate this model on the 36N library, which was obtained using a different strain and through different FACS thresholds, we used the 36N training set to refit the chemical potential and the hyperparameters of the logistic repression. Therefore, in the context of evaluating a model fitted on the $P_R$ library alone, both Validation and Evaluation 36N datasets can be considered true validation datasets.

Ultimately, we wanted to pool the training datasets of all three libraries to fit a unique model to give us the best set of parameters. In this case, all Validation and Evaluation datasets were data-naive. By doing this, we increased the inference power of our models, allowing us to fit pairwise (dinucleotide) interactions between nucleotides and the rate of RNAP clearance from the initial binding site.

## Standard thermodynamic model

In thermodynamic models of gene expression, the amount of protein is directly proportional to the fraction of time RNAP is bound to the promoter sequence. In the simplest case, where RNAP can be either bound ('on') or not bound ('off'-state), the probability of the on-state is given by the formula:

$$P_{on} = \frac{1}{1+e^{(E-\mu)/k_B T}} \tag{1}$$

where $E$ is the binding energy, and $\mu$ the chemical potential of RNAP in the cytoplasm. All our experiments were performed at room temperature, thus $kBT = 0.59$ kcal/mol. For simplicity, we define energy and chemical potential in those units, and drop $kBT$ from subsequent formulas.

We further expressed binding energy as an independent sum of local interactions between RNAP and individual base pairs in the promoter region, with each of the four possible bases (A,C,G,T) contributing differently depending on the position in the promoter. Such $4 \times l$ matrix is referred to as the binding energy matrix. The values of the energy matrix are determined up to an arbitrary offset per position. We set the zero value as the binding energy of the wildtype $P_R$ sequence.

In the case when the promoter sequence $L$ is longer than the binding matrix $I$, the number of states is $N = L - l + 1$, and the *on*-probability takes the form

$$P_{\mathrm{on}} = \frac{\sum_{i=1}^{N} e^{-(E_i - \mu)}}{1 + \sum_{i=1}^{N} e^{-(E_i - \mu)}} \tag{2}$$

The Standard model assumes that only one element (i.e. binding site) in the sum dominates, and we follow other studies (*Bintu et al., 2005*; *Kinney et al., 2010*) when formulating our standard thermodynamic model as

$$P_{\mathrm{on}} = \frac{1}{1 + e^{\min(E_i) - \mu}} \tag{3}$$

### Toward the Extended model: spacer flexibility

Based on the common observation that RNAP has a flexible spacer length, we extended the Standard model by allowing for spacer lengths that differ from the canonical by up to 2 bp. This effectively increased the number of possible configurations by a factor of 5. The key difference that we introduced was to assign an energy penalty for each non-canonical configuration. Increasing the spacer flexibility beyond ±2 bp did not yield further benefits to predictability. Note that it is in principle possible that the $\sigma^{70}$-RNAP energy matrix depends on spacer length, so that a different energy matrix would need to be inferred for each spacer length. Exploring this possibility was beyond the scope of this project, as the experimental datasets would be too small for reliable inference if conditioned on individual spacer lengths.

### Toward the Extended model: cumulative binding

In order to account for multiple RNAP binding configurations (different positions along the promoter and different spacer lengths) that can lead to a productive transcript, we saw the necessity of performing the thermodynamic sum in *Equation 2* instead of just extracting the dominant binding as was done for the Standard model. To do this, we assume that $\sigma^{70}$-RNAP concentration is not high enough for two molecules to be simultaneously present at the promoter before they initiate transcription. Fully embracing the thermodynamic description at this point provided us with a natural language for all further extensions.

### Toward the Extended model: occlusive unproductive binding

We accommodated the occlusive unproductive binding states naturally in the thermodynamic description:

$$P_{\mathrm{on}} = \frac{\sum_{i \in \{on\}} e^{-(E_i - \mu)}}{1 + \sum_{i \in \{np\}} e^{-(E_i - \mu)} + \sum_{i \in \{on\}} e^{-(E_i - \mu)}} \tag{4}$$

The main challenge here was to find the exact position after which the transcripts that did not contain a full (i.e. non-functional) RBS were produced. To answer this question, we checked computationally whether there was a natural region where this would be the case. First, we aligned all sequences with respect to the position of the RBS (Shine-Dalgarno sequence 'AGGAG'). Then, using the parameters from the previous model iteration (with cumulative binding), we assessed performance with different positions separating the productive and unproductive binding. This identified an 8-bp-long region, where the separating position was to be expected (*Figure 2B*, *Figure 2—figure supplement 2B*). To determine the location more precisely, we set up a separate experiment (see section *Verifying promoter structural features*).

## Toward the Extended model: occlusive binding on the reverse complement

When bound to the reverse complement, we considered RNAP to effectively act as its own repressor. We accommodated this in the model by expanding the set of 'np' states in *Equation 4* to all reverse complement configurations. Including this effect led to a slight but significant increased performance even without refitting, after which we also re-optimized the parameters locally.

## Toward the Extended model: dinucleotide interactions

The previous components of the model were fit exclusively on the $P_R$ dataset. In our search for dinucleotide interactions and when exploring departures from thermodynamic equilibrium, we pool the datasets in search for the best fit, as only then could we get a significant estimate of the interactions.

A common assumption of thermodynamic modeling of gene expression is that the binding energy of any particular transcription factor-promoter complex can be expressed as an independent sum of interactions between individual base pairs and the transcription factor (RNAP in our case) residues. A naive inclusion of all possible dinucleotide interactions between the contact residues of RNAP and the promoter would inflate the number of parameters to thousands, rendering their simultaneous estimation hopeless, even before considering overfitting issues.

To overcome the pitfalls of overfitting and to estimate the potential importance of dinucleotide interactions, we first included each interaction independently, and required that the best fit increased log-likelihood by at least 2-fold. This reduced the number of interactions from 4416 to 892. Then, we drew 10 random subsets of 20 interactions, and jointly optimized the interactions in the vicinity of the previously obtained values, over 10 cross-validation splits (50:50) of the training and validation datasets and using L1 regularization. This way, we obtained estimates (and associated error) for the interaction value in 10 different 'backgrounds'. For each interaction, we considered valid only those estimates that were larger in magnitude than a defined threshold, and combined them into a single estimate (mean and standard deviation), allowing for a small amount of leniency due to rather strict L1 regularization. We chose $0.002k_BT$ as the threshold for acceptance and made sure the results were robust for thresholds of 0.001 and 0.005. We then filtered for only those interactions that were non-compatible with zero at confidence level of $2\sigma$. This brought down the number of interactions to 250. In the next step, we again jointly inferred subsets of interactions: this time 12 subsets of 50 interactions. For each interaction, we then estimated the single mean and standard deviation from the 12 values, and conservatively selected only those that were inconsistent with zero at more than $3\sigma$, which brought down their number to 77. Finally, we sequentially include dinucleotide interactions starting with the most and moving toward the least significant, requiring that each contributed to at least $3\sigma$-level in the background of all accepted up to that point. This left 31 interactions (*Figure 2D*).

## Toward the Extended model: clearance rate

Finally, we implemented our framework even closer to the real biology by noting that RNAP is not like any other DNA binding protein, in-so-far that it is in fact *not* in thermodynamic equilibrium: to make the transcript, it needs to leave its original binding site, that is, if it binds too strongly it cannot make the transcript. We model this by introducing a parameter $R$, as a rate with which RNAP is cleared away from the binding site, relative to the rate with which it is bound, in the limit of very strong binding. It effectively sets up an upper limit on the amount of time RNAP persists on the binding site, and should make the difference only with strong binding sites.

Consider a system where RNAP can exist in only three states: (i) bound at a productive position, (ii) bound at an unproductive position, and () unbound ('off'); and consider the transition rates among them (*Scheme 1*):

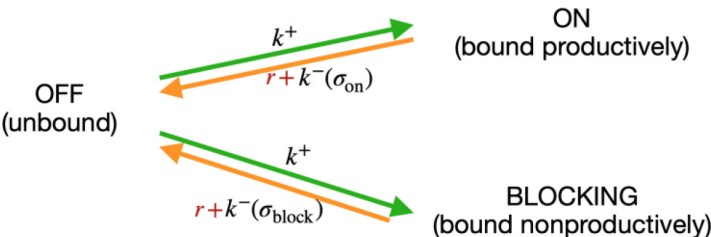

**Scheme 1.** Modelling RNA polymerase binding.

The transition rate from unbound to any of the bound states $k^+$ is the same and depends only on the concentration of free RNAP in the vicinity of the sites. The reverse transitions depend on the sequence ($\sigma$), through energy of binding $k^-_{\text{on(b)}} \propto e^{E_{\text{on(b)}}}$. We modeled RNAP clearance as an independent rate $\Gamma$, which depletes the bound states *on* and *b*, and eventually contributes to repopulation of RNAP in the cell. We do not model time dependence, and are interested only in the stationary state, thus we do not need to model the time delay between leaving the bound state and reappearing in the cell.

In stationarity we have:

$$\left(r + k^-_{\text{on}}\right) P_{\text{on}} = ck^+ P_{\text{off}} = \left(r + k^-_{\text{b}}\right) P_{\text{b}} \tag{5}$$

Introducing a relative clearance rate $R = \frac{r}{ck^+}$, casting the known rates and RNAP concentrations in terms of binding energy $\frac{k^-}{ck^+} = e^{\{E - \mu\}}$, and considering that $P_{\text{on}} + P_{\text{b}} + P_{\text{off}} = 1$, we obtained:

$$P_{\text{on}} = \frac{\left(R + e^{\Delta E_{\text{on}} - \mu}\right)^{-1}}{1 + \left(R + e^{\Delta E_{\text{on}} - \mu}\right)^{-1} + \left(R + e^{\Delta E_{\text{b}} - \mu}\right)^{-1}} \tag{6}$$

This is the same formula as Equation 4, with $e^{-(E_i - \mu)} \to \left(R + e^{\{E_i - \mu\}}\right)^{-1}$. Considering the clearance rate effectively introduced a cutoff on the binding energy $E_{\text{min}} = \mu + \log R$, below which the energy of binding is not important anymore. Note that in the case of $R \to 0$, the formula reduces to the previous state. In the case of a single (*on*) binding site, $R$ is a completely degenerate degree of freedom, which can be manifestly removed by a transformation $\mu \to \mu - \log(1 + R)$.

The scan over the values of $R$ (only on the training dataset) is shown in *Figure 2—figure supplement 2B*. For each value, we refit the chemical potentials. We obtained the best fit value for the relative clearance rate of $R^* = 0.31$, and a large uncertainty region, yet the model quite robustly preferred a non-zero clearance rate. Here, we consider the clearance rate to be a property of the RNAP molecule itself, and independent of sequence. Including the sequence dependence for the clearance rate can be done, however disentangling the exact nature of such dependence would require a dedicated set of experiments that is beyond the scope of this work.

## Verifying promoter structural features

We incorporated several structural features of bacterial promoters into the thermodynamic modeling framework (*Figure 1B*). When estimated on all three mutant libraries, each of these features independently increased model predictability, providing evidence of its biological role. Nevertheless, we conducted additional, hypothesis-driven experiments to verify the role of these structural features. For all mutant measurements performed for these validation experiments (outlined below), we measured expression levels of three biological replicates of each mutant in a plate reader. The mutants, the $P_{\text{R}}$ wildtype, and the negative control (the strain carrying the plasmid without a fluorescence marker in order to define 'measurable expression' – i.e. background fluorescence) were grown overnight in M9 minimal media supplemented with 0.2% (w/v) CAS, 0.2% (w/v) glucose, and 50 µg/mL kanamycin. The overnight populations were diluted 1000-fold, grown to $\text{OD}_{600}$ of ~0.1, and their fluorescence measured in a Bio-Tek Synergy H1 plate reader. All measurements were masked from the experimentalist performing them during the data allocation step. All the mutants were generated on the wildtype $P_{\text{R}}$ plasmid backbone, using oligonucleotide cloning. We did not conduct tests for the variable spacer, because changes in spacer length in $\sigma^{70}$-RNAP binding are well documented, so we assigned

them their accurate biophysical meaning in the form of a binding energy penalty to altering spacer length away from the lowest energy spacer (9 bp, corresponding to 17 bp between –10 and –35 sites). We did not independently verify the clearance rate, as we use a simplified description of clearance rate that attempts to capture its effects in a single parameter, rather than explicitly accounting for the mechanisms of σ70-RNAP clearance.

## Cumulative binding

To test for the role of multiple independent σ70-RNAP binding on gene expression levels, we created 29 specific promoters, which were derived from the $P_R$ wildtype promoter. These promoters were selected so that the prediction of their gene expression levels would be different between two models: (i) the Standard model, which did not include cumulative binding, predicted no measurable expression for these promoters; (ii) a model that included only cumulative binding as a structural promoter feature, and which predicted measurable expression. We measured gene expression levels of these promoters in a plate reader as described above, compared the expression to the negative control through FDR-corrected *t*-tests, and found that the model with cumulative binding described experimental observations systematically better (*Figure 2A*).

We introduced additional mutations into a subset of these promoters in order to remove the additional σ70-RNAP binding sites. To do this, we used the Extended model to determine which mutations in the 29 promoters would reduce binding to any but the strongest binding site. Finding mutations that removed the secondary binding site(s), but that do not affect the binding to the strongest σ70-RNAP binding site, was possible for only 7 of the 29 promoters. Removal of the additional binding site(s) in this manner led to a reduction of gene expression levels, and six of the seven mutants exhibited no measurable expression – that is, what was predicted by the Standard model which accounted for only a single binding site (*Figure 2A*).

## Occlusive unproductive binding

We performed several different tests to validate various aspects of this structural feature. First, we created 20 promoters to compare predictions from two models: (i) model that allowed for cumulative binding, but all binding had a positive impact on expression; (ii) model that allowed for cumulative binding, but considered every binding that would not transcribe a complete RBS as having a negative effect on expression. Model (i) predicted all 20 sequences to have measurable expression, while Model (ii), which accounted for occlusive unproductive binding, predicted no measurable expression for any of the sequences. By comparing the measured expression of each promoter to the negative control through FDR-corrected *t*-tests, we found that only 2/20 mutants exhibited measurable expression, in better agreement with the model that accounted for occlusive unproductive binding (*Figure 2—figure supplement 1B* top left panel).

Second, we wanted to test if our model accurately predicted how expression levels change when an additional occlusive binding site is introduced. To this end, we started with a wildtype $P_R$, and used the Extended model to generate a series of single point mutations that were predicted to gradually introduce a new binding site. σ70-RNAP binding to the newly introduced sites would result in transcripts without a complete RBS. By constructing these mutants in the lab, we found that introduction of such a site into the $P_R$ promoter indeed led to a significant reduction in expression levels (*Figure 2B* top right panel). Note that the strongest σ70-RNAP binding in the wildtype $P_R$ promoter is 28 bp upstream of the RBS.

We also did the reverse – starting with three promoters from the $P_R$ mutant library that had a strong occlusive unproductive binding site, we used the Extended model to predict a series of single point mutants that would gradually remove this binding site. Experimental measurements of those mutants, when compared to the expression level of the starting promoter sequence through FDR-corrected *t*-tests, exhibited a significant increase in the expression levels as the binding site was removed – an increase that was consistent with that site acting in an occlusive manner, and not a cumulative one (*Figure 2B* bottom left panel).

Finally, we wanted to determine the exact distance between the –10 element and the RBS that turned an additional binding site from being productive (cumulative) to being occlusive. To do this, we started with the same three promoters from the $P_R$ mutant library that had a strong occlusive unproductive binding site as above, but this time we shifted the occlusive position of that binding

site relative to the RBS. We shifted the binding site up to two positions closer to the RBS and up to seven positions further away from it, moving the binding site one position at a time – creating nine mutants of each of the three starting promoters. We used FDR-corrected *t*-tests to compare the measured expression of each mutant to the original sequence they were mutated from. We found support for the hypothesis that, as the occlusive binding site was moved further away from the RBS, it became cumulative (*Figure 2B* bottom right panel). This position matched the area determined with the model (see section *Toward the Extended model: occlusive unproductive binding*).

### Occlusive binding on the reverse complement

In order to conduct an independent verification of this structural feature of promoters, we identified four promoters from the 36N mutant library that the Extended model predicted to have strong occlusive binding on the reverse complement. We introduced up to eight mutations into each of these four promoters that would progressively eliminate the occlusive site on the reverse complement. The introduced mutations had minimal predicted effect on binding on the productive strand (*Figure 2C*, *Figure 2* – Extended *Figure 1*). We measured gene expression levels of these mutants as described above and performed a linear regression in order to correlate the measurements with the Extended model predictions of expression. For two sets of mutants, we found that the Extended model, which accounted for occlusive binding on the reverse complement, accurately predicted gene expression levels. For the other two mutant sets, removing the predicted binding site on the reverse complement had no measurable effect on expression levels (*Figure 2C*, *Figure 2—figure supplement 1*). This data shows that occlusive binding on the reverse complement is a more complex promoter feature than what we accounted for in the Extended model. Nevertheless, including this promoter structural feature into the model led to a significant increase in predictability (especially of the 36N dataset), which justified its inclusion into the Extended model.

### Verifying model predictions

Arguably the most surprising prediction arising from the Extended model is the ease of generating promoters from random non-expressing sequences. Specifically, we wanted to verify that single point mutations on random non-expressing 115-bp-long sequences could generate dramatic shifts in expression levels. To do this, we experimentally created 20 pairs of promoters, each pair consisting of (i) a random non-expressing sequence; and (ii) the same sequence but with one point mutation that is predicted to lead to expression. We created these promoters on the $P_R$ plasmid background and measured gene expression levels as described in the section *Verifying promoter structural features*. By conducting a series of FDR-corrected, paired *t*-tests we found that, indeed, single point mutations improved gene expression levels for all but two of the 20 promoters (*Figure 2—figure supplement 2A*). Of the 20 original promoters, which were predicted not to have any expression, only two exhibited expression (*Figure 2—figure supplement 2A*), confirming the accuracy of the Extended model on sequences of 115 bp length.

### Testing model generalizability with published datasets

In order to validate the generalizability of the Extended model, we tested how well it predicts gene expression levels obtained from *RNA-seq* experiments. To do this, we selected three published, large-scale, promoter mutant libraries whose expression was measured in *E. coli* (*Johns et al., 2018*; *Urtecho et al., 2019*; *Hossain et al., 2020*). These libraries had different properties in terms of how they generated promoter mutants and variants (as discussed in the main text).

Each library consisted of a promoter sequence matched to its measured gene expression level. For each library, we refitted only a single parameter in the Extended model – the chemical potential. All other parameters were fitted only on our three libraries ($P_R$, $P_L$, and 36N). We calculated *Pon* as a proxy for gene expression level from each promoter in the three tested libraries, and correlated our model prediction to experimental measurements to obtain the correlation coefficient ($r^2$). For the purposes of comparison, we did the same for the Standard model. This approach meant that the Extended model's predictability would very likely increase, if its parameters were to be refitted for each individual dataset. Furthermore, we did not include any genetic context around each promoter. In other words, we evaluated the Standard and the Extended models based only on the actual promoter sequence.

## Evolution simulations

We chose 100 random starting sequences of length 115 bp, so that their predicted expression is less than measurable under both the Extended and the Standard model. We implement Gillespie-type simulation under the assumptions of Strong Selection and Weak Mutation, that is, when we can assume no clonal interference. We define the time scale in the units of inverse mutation rate $1/\mu$, so in each iteration of the algorithm we simulate a single mutation that appears and gets fixed in the population with the probability given by the Kimura formula

$$P_{\text{fix}} = \frac{1-e^{-2s\delta\phi}}{1-e^{-4Ns\delta\phi}}$$

where $N$ is the population size, $\delta\phi$ is the change in the phenotypic value (gene expression level) due to mutation, and $s$ is the selection strength on the phenotype. Note that phenotype and selection are degenerate parameters: only their product represents the actual selection coefficient. We postulate $\phi = \log_{10}P_{\text{on}}$, so the typical mutational effects are in the range of $10^{-3}$ to $10^{-2}$. Under this regime, selection strength of 100 means a selection coefficient in the range 0.1–1.

We went through a sufficient number of iterations (new mutation events) until a threshold expression was reached by the population. We considered two thresholds: measurable expression we identified in our experiments to model the emergence of novel (weak) constitutive promoters; and the expression of the wildtype Lambda $P_R$ promoter, as an example of a very strong promoter with high constitutive expression levels. For each of the 100 starting sequences, we performed 100 independent evolution runs, to obtain an estimate of the mean exit time (and the associated standard error of the mean) under the Standard and Extended models.

We also varied the number of nucleotides in the 115-bp-long sequence that were allowed to mutate. The mutagenized region was always in the center of the sequence, over which we always integrated to obtain expression. In simulations with a more constrained region that was allowed to mutate, the simulation would take too long until full convergence. To be robust against these cases, we stopped each simulation after 10N steps, truncating the distributions of evolution times for those sequences. Importantly, doing this introduced a bias that affected the Standard model more so than the Extended (because the Standard model took longer to reach the threshold), leading to an underestimate of how much faster the Extended model is compared to the Standard one. As such, what we report as the increase in the rates of evolution of the Extended model compared to the Standard one is likely the lower bound.

## Insights from applying the model to the *E. coli* genome

For all analyses described in this section, we used the *E. coli* K12 MG1655 genome (NCBI reference: NC_000913). Based on the genome annotations, we assigned one of two identities for each of the nucleotide positions in that genome: (i) 'within genes' (intragenic) – if the position was a part of the following annotation types: CDS or gene; (ii) inter-genic – if none of the following annotation types were present at the position: misc_feature, mobile_element, repeat_region, tRNA, Sequence Tag Sites (STSs), tmRNA, rRNA, CDS/gene, ncRNA. 'Within genes' region accounted for 89.6% of the genome with 51.8% GC content, while 'inter-genic' regions formed 9.6% of the genome with the GC content of 41.1%. Because the inter-genic category defined in this way does not guarantee that the sequence actually has a promoter function, we also considered only those sequences that have been experimentally identified as promoters, as found on RegulonDB.

We calculated binding energy for each of the 5× $G$ configurations (five spacer lengths, $G$ is the number of positions/base pairs in the genome), and then performed a thermodynamic sum over all spacer lengths for each position aligned so that each configuration coincided with the last base pair of the −10 element of the binding matrix. This way we obtained a free binding energy for each nucleotide (for each of the $G$ positions) in the genome. We offset all the obtained free binding energy values such as to set the minimum to zero, in the interest of later easier readability. Surprisingly perhaps, the nearest-neighbor correlation of the free energy values is negligible, so we did not subsample, but kept all values.

To assess whether there was selection against $\sigma^{70}$-RNAP binding sites, we constructed a synthetic genome of 100 million base pairs with the same GC content as that region (inter-genic or within-genes) of the *E. coli* genome. We histogram the cumulative distributions, and normalize them to

the number of base pairs (inter-genic or within-genes) relevant for the real *E. coli* sequence. For the p-value plot, we calculated the cumulative mass function of the Poisson distribution, at the value of the real histogram, and with mean given by the synthetic histogram value. This corresponded to the one-tailed p-value.

We also developed an alternative method of evaluating selection against $\sigma^{70}$-RNAP binding sites, in order to strengthen the validity of our claims. This time, instead of creating a random synthetic genome and comparing the predicted expression levels across that genome using the $\sigma^{70}$-RNAP energy matrix (*Figure 1C*), we created 100 shuffled energy matrices and evaluated free energies of such models across the actual *E. coli* genome. Matrices were permuted *per* position, meaning that the columns of the matrix were shuffled without altering the internal structure of each column. For each such matrix, we evaluated the model at every single position in the *E. coli* matrix and calculated the cumulative histogram, as in the previous paragraph. The p-values were calculated assuming a normal distribution per bin. This is in fact a conservative estimate of the p-value, as for those bins with lower means, Gaussian overestimates the variance and hence the p-value. In the plot we illustrate the 95% confidence interval by showing explicitly the 3rd and the 97th percentile.

To predict the expression from known *E. coli* promoters, we assigned a free energy to each of the 1951 promoters in RegulonDB by integrating over a symmetric 40 bp region around the reported transcript start. For a fair comparison in plot, we also integrated in the same way over 40 bp chunks of all the inter-genic regions described above (red line in *Figure 4F*). For each promoter, we then searched through RegulonDB for all transcription factors that bind it. If all of them were activators, we flagged that promoter as 'activatable', and similarly for 'repressible'. If we found a mixture of activators and repressors affecting a given promoter, we flagged the promoter as 'dual'. The promoters with no known transcription factor binding were flagged as 'no info'.

## Acknowledgements

We thank Hande Acar, Nicholas H Barton, Rok Grah, Tiago Paixao, Maros Pleska, Anna Staron, and Murat Tugrul for insightful comments and input on the manuscript. This work was supported by: Sir Henry Dale Fellowship jointly funded by the Wellcome Trust and the Royal Society (grant number 216779/Z/19/Z) to ML; IPC Grant from IST Austria to ML and SS; European Research Council Funding Programme 7 (2007–2013, grant agreement number 648440) to JPB.

## Additional information

### Funding

| Funder | Grant reference number | Author |
| --- | --- | --- |
| Royal Society / Wellcome Trust | 216779/Z/19/Z | Mato Lagator |
| European Research Council | 648440 | Jonathan P Bollback |
| IPC Grant from IST Austria | | Mato Lagator Srdjan Sarikas |

The funders had no role in study design, data collection and interpretation, or the decision to submit the work for publication.

### Author contributions

Mato Lagator, Conceptualization, Investigation, Methodology, Writing – original draft, Writing – review and editing; Srdjan Sarikas, Conceptualization, Formal analysis, Methodology, Writing – review and editing; Magdalena Steinrueck, Conceptualization, Investigation, Methodology; David Toledo-Aparicio, Investigation, Writing – review and editing; Jonathan P Bollback, Writing – review and editing; Calin C Guet, Conceptualization, Investigation, Methodology, Writing – review and editing; Gašper Tkačik, Conceptualization, Formal analysis, Investigation, Writing – review and editing

## Author ORCIDs

Mato Lagator (ID) http://orcid.org/0000-0001-7847-3594
Magdalena Steinrueck (ID) http://orcid.org/0000-0003-1229-9719
David Toledo-Aparicio (ID) http://orcid.org/0000-0001-8381-7590
Jonathan P Bollback (ID) http://orcid.org/0000-0002-4624-4612
Calin C Guet (ID) http://orcid.org/0000-0001-6220-2052
Gašper Tkačik (ID) http://orcid.org/0000-0002-6699-1455

## Decision letter and Author response

Decision letter https://doi.org/10.7554/eLife.64543.sa1
Author response https://doi.org/10.7554/eLife.64543.sa2

## Additional files

### Supplementary files
• Transparent reporting form

### Data availability

Source data files have been provided for Figures 1 & 3. Code and data has been deposited in GitHub: https://github.com/szarma/Thermoters copy archived at swh:1:rev:61fe2f54941966469dad801efe06e1c879f27530.

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
