## [Editor Report]

This paper builds a biophysical model to understand the contribution of the DNA sequence in σ70-RNA polymerase binding activity. The authors provide evidence that taking into account that RNA polymerase can bind in multiple configurations, including non-productive ones, significantly improves predictions. They also confirm and extend previous observations that functional promoter sequences are relatively abundant in random sequences. This work represents an important advance to the field, and will hopefully serve as the stepping stone for better models that include transcription factors and eukaryotic enhancers.

---

## [Decision Letter]

**Decision letter after peer review:**

Thank you for submitting your article "Predicting Promoter Function and Evolution from Random Sequence" for consideration by *eLife*. Your article has been reviewed by 3 peer reviewers, and the evaluation has been overseen by a Reviewing Editor and Aleksandra Walczak as the Senior Editor. The following individuals involved in review of your submission have agreed to reveal their identity: Erik van Nimwegen (Reviewer #3); Tal Einav (Reviewer #4).

The reviewers have discussed the reviews with one another and the Reviewing Editor has drafted this decision to help you prepare a revised submission.

While the reviewers found that the manuscript contains potentially interesting results, as you can see from the reviews appended below, they have many concerns. Most importantly, they are not fully convinced that the model accurately predicts expression levels and would like to see much more analysis of which extensions of the previous thermodynamic models are important for the accuracy of such predictions. They were also concerned that the result that promoters are common among random sequences is not novel and does not warrant the emphasis that the manuscript puts on this result. Therefore, we suggest that in a revised manuscript you:

(i) focus primarily on the claim that the extensions to the previous thermodynamic model are important for explaining the expression levels from random sequences and real promoters, and show in greater detail how each extension is supported by instances in your dataset;

(ii) quantify more explicitly the accuracy of predictions from your model following the suggestions of the reviewers, and specifically quantify the importance of the extensions to previous models for predicting the expression levels in your libraries and in native constitutive promoters;

(iii) test your model on at least one (preferably more) of the additional data sets suggested by Reviewer 4, and provide a clear summary of how your results compare with the data.

Please do also go through the reviews below and try to address all the additional points raised by the reviewers.

*Reviewer #2:*

This paper builds a biophysical model to understand the contribution of the DNA sequence in σ70-RNA polymerase binding activity. To train their model, which takes into account intra-motif spacing as well as dependencies, they generate large mutant promoter libraries and measure expression from those. The title is a bit too general since the data and model are for a specific constitutive bacteriophage promoter, but nevertheless, the study has novel implications in evolution of gene regulation.

Lagator et al., have modeled the binding of σ70-RNA polymerase (RNAP) to a constitutive promoter to understand the effect of the promoter sequence in regulating gene-expression. This is a biophysical model that extends the standard thermodynamic model of RNAP binding with its cognate -35 and -10 boxes in several ways: the aim is to account for a variable spacer between the two boxes, binding at non-cognate regions, hindering transcription by binding on the negative strand (opposite orientation), occlusive binding (causing the transcript to begin upstream and not contain the RBS), and relaxing the assumption of independence within the energy matrix by modelling pair-wise nucleotide dependencies. They create three mutant libraries originating from strong λ pR and pL promoters as well as a completely random (25% probability of each nucleotide) set of oligonucleotides to train their model parameters. The model shows an impressive predictive power on a held out set and predicts expression patterns of completely random sequences to a remarkable degree.

They further conclude that many non-promoter *E. coli* regions are only one bp mutation away from giving rise to measurable expression. They validate their predictions on promoters where the standard model predicts a different expression pattern from their model. The data from their experiments will be a valuable addition to the community in case people want to develop new models or test other hypotheses.

A few things are unclear:

1. The thermodynamic model still makes some assumptions which need to be clearly mentioned. The sum suggests there is at the most, a single RNAP molecule (or a binding site at a time) present on each promoter.

2. The authors learn the models by using multinomial logistic regression (MLR), where the response variable is the median observed expression for each bin (which are four/12 in total, depending on the promoter class). This is a little unusual, since the four expression values are not "categories", which is typically the expectation in MLR. In other words there is a definite order (or dependence) within the categories, the first bin having a lower expression than the second, etc. That information is lost/unused in MLR.

3. It is not clear whether there are enough instances of spacer width varying in the mutant pR / pL libraries. If the mutations are single nucleotide changes (not deletions/insertions), would it not be highly unlikely that the library contains instances of the two boxes having different spacing?

It will be best to mention clearly the assumptions in the cummulative model, mathematically. A more detailed explanation of the sum in equation 2 will be useful. See for example, in the Bintu et al., paper cited in the same section. Similarly, the MLR model should be described with an equation.

Figure 1 C shows the learned energy matrix. There are other studies which use the standard model to learn their own matrix for RNAP from large mutant libraries. A comparison between the matrices will be useful, even if the promoters are different. In the same vein, this new model can be applied on those data to see if the expression can be better predicted. This will show the generality of the model.

Does sequence in pR (and pL) contain also the annotated transcription start site (TSS)? Is there any chance that the TSS also gets mutated in their libraries? Since the -10 box is relative to the TSS the distance between the box and TSS is presumably important, which can be defined only with respect to the nucleotide (or a small neighborhood) at the TSS. Does the final energy matrix contain those regions? Figure 1C suggests not.

I would have expected to see the Y axis of Figure 2B to be stratified across 12 bins. But that is not the case, which means I am missing something.

Is it not possible that the energy matrix values (those at the two boxes) change with the spacer? I understand that is not allowed in the current model.

Finally, the contribution of the dinucleotide interactions is not convincing. Perhaps the prediction accuracy on the held out set based on with and without including these terms will help clear that.

*Reviewer #3:*

Lagator et al., extend previous biophysical models to predict gene expression of constitutively expressing *E. coli* promoters from their sequence. They provide evidence that taking into account that RNA-polymerase can bind in multiple configurations, including non-productive ones, significantly improves predictions. They also confirm and extend previous observations that functional promoter sequences are relatively abundant in random sequences.

The aim of this study is to develop a realistic biophysical model for predicting the expression of (constitutive) *E. coli* promoters from their sequence. From my understanding of the presentation, the main new results that the authors claim are:

1. They extend existing biophysical models to a more realistic model that takes into account that the spacers between the -35 and -10 sites are flexible, that one should not just focus on the best site in a region but take into account cumulative binding to all possible sites, and that one should also explicitly model that binding to some sites is not productive (e.g. binding to reverse-complemented sites on the opposite strand).

2. They confirm experimentally that these extensions can indeed have a major impact on expression, for selected promoter sequences.

3. This biophysical model accurately predicts the gene expression level of any sequence.

4. They show that a substantial fraction of short random sequences will act as promoters, i.e. drive significant expression.

5. There is significant evidence for selection against ubiquitous expression, i.e. sigma70 binding sites are depleted both in gene bodies and in intergenic regions.

In my opinion the strongest parts of the paper are points 1 and 2. The extensions of the 'standard' biophysical model are all very sensible and the model is implemented in a mostly reasonable way. None of the ingredients in the model are particularly novel or surprising, but that's not a bad thing. I do wonder why the authors did not include features such as allowing only one of the two 'feet' of the sigma70 to bind (like in the Einav and Phillips PNAS 2019 paper) or non-specific binding (although maybe the clearing rate that the authors introduce plays a role analogous to non-specific binding). It also was not entirely clear to what extent binding at one site precludes binding at other sites in the model. This could be explained better. But apart from this the model is very reasonable. The only part I am sceptical about is the dinucleotide modeling, which is based on a very ad hoc and rather baroque procedure.

I particularly liked the experimental confirmations (Figure 1 – supplement 1) that support that several extensions of the model can indeed have a major impact on expression for selected promoter sequences. In my opinion these results should probably have been the main focus of the paper, maybe together with some more rigorous quantification of what role the various extensions play in predicting the expression levels of the random sequences. If, in addition, the authors had provided evidence that these extensions are also important for expression in native constitutive *E. coli* promoters, I think this could have been a very nice paper.

However, currently the focus seems to be on two aspects that I am much less convinced about. The authors in several places make the claim (point 3 above), that their model accurately predicts expression levels. I do not feel this claim is properly substantiated. First, to give some context, there is a very large body of work going back to the 1980s that aim to predict promoter sequences in *E. coli* directly from sequence, and with just a little bit of googling one will uncover plenty of works in the literature that appear to claim to `solve' this problem, e.g. Cassiano and Silva-Rocha mSystems 2020 present a recent benchmarking of a whole host of promoter prediction tools. I understand that these methods do not specifically aim to predict promoter strength but presumably one could use the 'score' that they calculate as a predictor of promoter strength and it is not clear to me how poorly such predictors would perform in comparison with the model presented here. In addition, there are also several previous papers that specifically claim to successfully predict promoter strength from sequence using either biophysical models close to the ones used here (e.g. Brewster et al., PLoS Comp Biol 2012, Einav and Phillips PNAS 2019) or more general machine learning methods, e.g. Mulligan et al., Nucl Acids Res 1984, Weiler and Wecknagel J Theo Biol 1994, DeMey et al., BMC Biotechnology 2007, Meng et al., Quantitative Biology 2017, and the recent Zhao et al., BioRxiv 2020.06.25.170365. Does the model of the authors significantly improve over these approaches? It is not clear to me. Especially the Einav and Phillips method is curtly dismissed in the discussion, but it is not clear to me whether that model, if trained on this data, would really perform much worse than the current model.

I understand it is kind of a pain to do extensive comparison with previous methods but in order to make real progress on understanding what determines the expression levels of constitutive *E. coli* promoters, then at a minimum I would have expected a very clear quantification of exactly how well the model performs. What one would want to know is: If I feed the model a random sequence, how accurately does it predict its expression level? Or, how well can the model predict the variation in expression levels of constitutive promoters across the *E. coli* genome? How much accuracy will be lost when various of the extensions are removed, either on native promoters, or on random sequences.

However, we do not really get such an analysis. The authors only present one scatter and Pearson correlations between the predicted log[Pon] and measured log[expression] for various versions of the model. Crucially, I do not think that a correlation of r^2^ ~ 0.75 on a dataset where expression levels vary over a 1000-fold range can be reasonably described as `accurate prediction'. Just as a back-of-the-envelope, if both data and model have a standard-deviation in log-expression levels of about log[100], then r^2^ = 0.75 corresponds to the error between model and data having standard-deviation of log[10]. That is, unless I made a mistake in my back-of-the-envelope, the model's predictions would typically be off by 10-fold (and the scatter does seem sort of consistent with this). t is also in line, with the fact that the bins for the P_R data are 10-fold apart in expression level, i.e. the training of the model is done on a dataset in which promoters whose expression levels are within 10-fold of each other are effective treated as equally expressed. I do not think that this level of precision can be called accurate prediction of expression levels. For example, using very similar reporter constructs, the difference in expression of the median and 95th percentile of native *E. coli* promoters is about 20-fold (i.e. see Wolf et al., Figure 1B), so this model could barely tell these apart.

Another issue is that it is not clear which extensions of the model are the most crucial for the accuracy of the predictions. I was surprised to see (in Figure 2 – source data 1) that, apparently, by far the biggest improvement on the random sequences (36N dataset) comes from including flexible spacers and that cumulative binding and occlusive binding adds rather little. This appears somewhat at odds with the narrative that the authors present in the paper that stresses the importance of cumulative binding.

In short, although I would like to believe that this reasonable biophysical model improves a lot over previous approaches in the literature, I don't think a convincing case has been made that this is really a major advance in accuracy of promoter strength prediction.

Finally, regarding points 4 and 5, I do not understand why the authors place so much emphasis on the observation that functional promoters (i.e. sequences that drive gene expression) are common among random sequences, because this is neither novel nor surprising. It has been long-known that the 'position specific weight matrices' that model sigma70 binding sites have low information content, e.g. Schultzaberger et al., Nucl Acids Res 2007 estimate it to be about 6.5 bits. Given this, one would very roughly expect a site every 100 bp in random sequences. In fact, this was one of the reasons why, in Wolf et al., (*eLife* 2015), we were confident that we would easily find a large diversity of functional promoters in a library of random sequences. This expectation was confirmed by our measurements as one can see from Figure 1B of Wolf et al. That is, after selecting the 5% of highest expressing cells from the random library, most of them express at a reasonable level. At the time it didn't even occur to us to mention this as a surprising observation. When the paper of Yona et al., (Nat Comm 2018) appeared, we realized that we misjudged this, and that for many people it WAS surprising that functional promoters are common even in relatively short random sequences. However, now that Yona et al., have an entire paper focused on this observation, including showing depletion of sigma70 sites within gene regions, there is no more novelty to this observation.

The observations about the frequency of occurrence of promoters in random sequences could have been more interesting if it had been more meaningfully quantified. That is, any sequence will presumably drive expression at some nonzero rate. An interesting (and meaningful) quantitative question is how likely it is that a random sequence drives expression at rate > x, as a function of x. Maybe the authors' data and model could give a reasonable answer to this question. However, instead the authors simply treat this problem as if there is just a binary question whether a promoter is expressing or non-expressing. Moreover, they define the threshold between expressing and non-expressing in a way that is not biologically meaningful but seems determined by instrument precision. I actually had a hard time understanding how precisely 'measurable expression' was defined. My understanding is that, for the P_R and P_L libraries, it is defined as any expression higher than the 99th percentile of measurements for cells carrying no YFP. But how it is defined for the 36N library (with another reporter) was unclear to me. In any case, it appears that 'measurable expression' is defined in terms of the auto-fluorescence level that the cells have in the FACS machine. Unfortunately, this is not a biologically meaningful cut-off. In fact, a substantial fraction of *E. coli* promoters express below the detection limit in the FACS (i.e. not significantly above auto-fluorescence).

In summary, I think there are some very good ideas in this paper. I in principle like the biophysical model, and the experimental validations of the model's extensions I thought were very nice and, when extended, could make a compelling paper. Unfortunately, the current focus of the papers appears to be on the frequent occurrence of promoters in random sequences (which is not novel) and claims that the model is a major advance in predicting gene expression from sequence, which I do not feel are substantiated by the results presented.

1. I had real trouble understanding the processing of the 36N dataset. I understand that, because in the 12-bin dataset you sort until you have a certain number of cells in the bin, you need to correct for this when estimating the relative fractions of cells with a given promoter that go to different bins. But I do not understand the approach with the spike-ins. Aren't there, in addition to the spike ins, also an unknown number of reference promoters coming from the sorting?

2. line 81: "even for constitutive promoters, where σ70-RNAP binding solely determines gene expression levels" Has this really been established? There could be effects of differential binding of topoisomerases that undo supercoiling.. or differential weak binding by other TFs.. and so on. I don't think we really know these levels are solely determined by sigma70-RNAP.

3. Figure 1 – supplement 1.

Why is detectability threshold higher in B than in A? I found it generally quite frustrating that it is hard to figure out how exactly this 'measurable expression' threshold, which plays such a crucial role, is defined.

4. 515-516: Finally, we required the distribution of expression bins to be as unambiguous as possible (Figure 2 —figure supplement 1D,E).

Why is this done? The way I understand it, this throws out promoters whose expression levels straddle two bins. I guess you want to throw those out because the fitting of the model to only observations in 4 bins only is more accurate when using only promoters that fall sort of in the middle of those bins, but I find this quite unsatisfactory. Would it not have been possible to fit a log-normal to the relative fractions of counts in the different bins and get continuous expression estimates?

5. 596: "We define 60:20:20% splits of each of our libraries into three disjunct datasets Figure 2 – Source Data 3."

Are these splits random? A notorious problem in cross-validation based fitting is when highly correlated datapoints occur in both training and test data. For example, in this case you could have promoters that are virtually identical, and with equal expression, in both training and test datasets. This could lead to misleadingly high performance on the test dataset. I actually doubt this is happening here, but maybe it is worthwhile checking this.

6. Did you try adding a non-specific binding term to the model? Or the ability for sigma70 to bind through only one of its two feet? If not, why not? Do the authors think sigma70 cannot be bound only at the -10 or -35 sites?

7. I did not like the way multinomial logistic regression was used to fit the model. Expression should go up monotonically with Pon. It makes no sense to treat the bins as unrelated 'classes'.

8. Comparing evolution of promoters de novo under the extended model against the 'standard model' is a straw man in my opinion. The standard model is an approximation to the thermodynamic model that is valid when there is one site that dominates the overall affinity of a promoter. I agree that it is often assumed that this is a good approximation for native promoters that have been shaped by selection. But I do not think there is anybody that thinks this is a good model for how promoters appear in random sequences. Obviously weak promoters can then appear anywhere in the sequence.

9. For Figure 3A, I think it is essential to also predict such a curve for the sequences in each of the libraries and see how well these predicted curves match the observed distributions for these libraries. Otherwise it is hard to judge how meaningful this predicted distribution is.

10. Many intergenic sequences in *E. coli* will be constrained to have all kinds of TFBSs, precluding sigma70 to occur in those positions. So to check whether there is really depletion of sigma70 sites, comparing with random sequence of the same nucleotide composition is not quite correct in my opinion. First, you should distinguish intergenic regions that are upstream of two operons from those that are upstream of one, and those downstream of two promoters. Those 3 classes of intergenic regions are also known to have different nucleotide composition. The best regions for tests would be downstream regions, and regions that are upstream of an operon that is constitutive.

*Reviewer #4:*

The manuscript by Lagator et al., examines a general framework to predict the constitutive level of gene expression for bacterial DNA, which constitutes one of the most fundamental processes in biology. Whereas previous work has typically been restricted to a limited sampling around a well-studied promoter (such as the lac operon), the authors push towards a model that can characterize all possible sequences, which they support with libraries containing 10,000 mutants each around two strong promoters (PR and PL) as well as a library containing 10,000 random sequences (36N). This work represents an important advance to the field, and will hopefully serve as the stepping stone for better models that include transcription factors and eukaryotic enhancers.

While the authors provide a clear and compelling story using their impressive data set, the one piece that I felt was missing was a connection with large gene expression data sets from other groups. How robust will their already-trained model be when applied in another context, where the experimental design is somewhat different? This will help other groups understand the potential gain in implementing this model for their own ends, and if any fine-tuning or model adjustments need to be performed, such information would be important to add. To this end, I suggest a few possible data sets below. The authors do not need to analyze all of them (although more is better), and they are welcome to substitute other data sets of their choice:

1) Hossain 2020 [https://doi.org/10.1038/s41587-020-0584-2] analyzed 4,350 bacterial promoters with a broad range of gene expression measurements varying by 820,000-fold. These promoters should be constitutive, although their random DNA sequences might occasionally give rise to transcription factor binding sites (as in the 36N library). The sequences and read counts are available in their Supplementary Data 3.

2) Urtecho 2019 [https://doi.org/10.1021/acs.biochem.7b01069] analyzed the constitutive expression from 10,000 promoters. Their setup ensures that no transcription factors are involved, providing a clean test for the model. You can find their expression data at the GEO Accession Number GSE108535 [https://www.ncbi.nlm.nih.gov/geo/query/acc.cgi?acc=GSE108535]. The file “GSE108535_sigma70_variant_data.txt.gz” contains the names of promoter variants and the average RNA/DNA expression levels of two biological replicates, while the file “GSE108535_barcode_mapping.txt.gz” contains the name of each promoter variant and its corresponding sequence (where the sequence is specified in the column ‘most_common’). Figure S12 and Table S1 from their supplementary information provide a visual schematic for how to put the pieces together.

3) Johns 2018 [http://doi.org/10.1038/nmeth.4633] analyzed 30,000 promoters mined from prokaryotic genomes, which would be a very interesting avenue to explore. These promoters may involve transcription factors (although their mining strategy might bias them towards constitutive promoters). Their sequences are given in Supplementary Table 1 and their expression measurements for *E. coli* are provided in Supplementary Table 3.

I suspect the authors are already planning to do this, but it would be very helpful if a program implementing the fully-trained model was provided (along with some examples of its usage), to help the community utilize these results.

---

## [Author Response]

While the reviewers found that the manuscript contains potentially interesting results, as you can see from the reviews appended below, they have many concerns. Most importantly, they are not fully convinced that the model accurately predicts expression levels and would like to see much more analysis of which extensions of the previous thermodynamic models are important for the accuracy of such predictions. They were also concerned that the result that promoters are common among random sequences is not novel and does not warrant the emphasis that the manuscript puts on this result. Therefore, we suggest that in a revised manuscript you:(i) focus primarily on the claim that the extensions to the previous thermodynamic model are important for explaining the expression levels from random sequences and real promoters, and show in greater detail how each extension is supported by instances in your dataset;(ii) quantify more explicitly the accuracy of predictions from your model following the suggestions of the reviewers, and specifically quantify the importance of the extensions to previous models for predicting the expression levels in your libraries and in native constitutive promoters;(iii) test your model on at least one (preferably more) of the additional data sets suggested by Reviewer 4, and provide a clear summary of how your results compare with the data.Please do also go through the reviews below and try to address all the additional points raised by the reviewers.

We thank the Editor and the Reviewers for their comments and suggestions. We have addressed each comment raised by the Reviewers (see below). In particular, we analyzed the performance of our model on three datasets suggested by Reviewer 4, and compared our model performance to that of the models presented in various papers, in order to demonstrate the cross-predictability of our model. We have also expanded on the description of different extensions we added to the existing thermodynamic model, and explored in more detail how each contributed to predictability.

However, while we appreciate that the Reviewers found the ‘evolutionary’ part of the manuscript less novel, we continue to strongly believe that it constitutes an important contribution to the field. This is because the question of whether expression emerges easily from random sequences is a *quantitative* – not a *qualitative* – one, as we argue below.

A few previous studies (Wolf, Silander, Nimwegen 2015 and Yona *et al.,* 2018) have shown that random sequences often lead to expression, but have not offered a quantitative understanding of the phenomenon. For example, Yona *et al.,* 2018, used a small dataset of only 40 sequences to show that ~10% of them have promoter activity. The small dataset inherently prevents one from drawing quantitative conclusions about just how likely random sequences are to lead to expression. To obtain a more quantitative understanding of this problem, one could use a standard energy matrix (such as the one obtained by Kinney *et al.,* 2010) over random sequences and identify the likelihood of random sequences leading to expression. This is, in fact, exactly what we did in this study by using the Standard model, which represents a conservative *null* expectation for our studies.

In contrast to this null expectation, we find that the Extended model, which accounts much better for the experimental data, also predicts a significantly higher likelihood of random sequences leading to expression. This quantitative (as opposed to qualitative) understanding of σ^70^-RNA polymerase binding also allows us to make more precise statements about how promoters evolve from random (and non-random) sequences, allowing us to claim not only that promoter evolve rapidly, but that they evolve *much more* rapidly than previously thought. Similarly, we identified *stronger* selection against σ^70^-RNAP binding sites than was previously thought, as we find evidence of selection against binding sites even within promoter sequences (previous studies have reported selection against binding sites within coding regions only). Importantly, these effects are not small: the fraction of sequences that express above a detectability level is more than double using the Extended compared to Standard model, and the predicted rates of evolution of functional promoters can be faster by orders of magnitude (!) using the Extended compared to the Standard model.

Put together, our study achieves something unique: it uses a large-scale dataset to develop a model with higher predictability compared to state-of-the-art mechanistic models of gene expression, and then uses that model to improve our quantitative understanding of promoter evolution. In doing that, it points to a potentially huge effect that molecular details of binding of a protein (RNAP) could have on the evolvability of gene expression.

We have made numerous changes to the text to reflect and better emphasize what is novel, and in particular to emphasize the fact that we predict promoter evolution to be not “just” fast in general, but indeed much *faster* than what the previous state-of-the-art models would predict. We elaborate further on why we think this evolutionary aspect of our study is important in responses to the individual points raised by the Reviewers. The lines referenced in this letter correspond to the revised manuscript document with tracked changes.

Finally, we would like to apologize to the Editor and the Reviewers for the long time it took us to address all the comments. A combination of the co-authors living in different locations and changing work places (and career paths in some cases), as well as the Covid-19 pandemic introduced substantial delays.

Reviewer #2:This paper builds a biophysical model to understand the contribution of the DNA sequence in σ70-RNA polymerase binding activity. To train their model, which takes into account intra-motif spacing as well as dependencies, they generate large mutant promoter libraries and measure expression from those. The title is a bit too general since the data and model are for a specific constitutive bacteriophage promoter, but nevertheless, the study has novel implications in evolution of gene regulation.

We thank the Reviewer for the positive assessment of our manuscript. We have amended the title to be more specific, clarifying that we focus on bacterial promoters.

Lagator et al., have modeled the binding of σ70-RNA polymerase (RNAP) to a constitutive promoter to understand the effect of the promoter sequence in regulating gene-expression. This is a biophysical model that extends the standard thermodynamic model of RNAP binding with its cognate -35 and -10 boxes in several ways: the aim is to account for a variable spacer between the two boxes, binding at non-cognate regions, hindering transcription by binding on the negative strand (opposite orientation), occlusive binding (causing the transcript to begin upstream and not contain the RBS), and relaxing the assumption of independence within the energy matrix by modelling pair-wise nucleotide dependencies. They create three mutant libraries originating from strong λ pR and pL promoters as well as a completely random (25% probability of each nucleotide) set of oligonucleotides to train their model parameters. The model shows an impressive predictive power on a held out set and predicts expression patterns of completely random sequences to a remarkable degree.They further conclude that many non-promoter *E. coli* regions are only one bp mutation away from giving rise to measurable expression. They validate their predictions on promoters where the standard model predicts a different expression pattern from their model. The data from their experiments will be a valuable addition to the community in case people want to develop new models or test other hypotheses.A few things are unclear:1. The thermodynamic model still makes some assumptions which need to be clearly mentioned. The sum suggests there is at the most, a single RNAP molecule (or a binding site at a time) present on each promoter.

The Reviewer is correct to point out that our ansatz assumes that only a single RNAP molecule can be present at the promoter at a given time in a productive configuration. Once bound, the RNAP molecule initiates transcription and leaves a promoter faster than another RNAP molecule can find and bind to the promoter. In other words, we assume that RNAP concentration is not high enough for two molecules to be bound to the promoter simultaneously before they initiate transcription and clear the promoter. We explain this assumption in the main text (L117-119) and in the Methods (L804-806).

2. The authors learn the models by using multinomial logistic regression (MLR), where the response variable is the median observed expression for each bin (which are four/12 in total, depending on the promoter class). This is a little unusual, since the four expression values are not "categories", which is typically the expectation in MLR. In other words there is a definite order (or dependence) within the categories, the first bin having a lower expression than the second, etc. That information is lost/unused in MLR.

We implemented one-vs-rest MLR as a conservative choice, which would allow for a non-monotonic relationship between P_on and gene expression. Ordinal logistic model would force it by definition. In particular, for all the models without the clearance rate, if there were a binding state so strong that the expression is actually lower than for a bit weaker binding, we would be able to detect it (and we were actively looking if any such effect exists; it does not). Despite using MLR, we find a monotonic relationship between P_on and expression, which we took to be one of our internal consistency checks.

3. It is not clear whether there are enough instances of spacer width varying in the mutant pR / pL libraries. If the mutations are single nucleotide changes (not deletions/insertions), would it not be highly unlikely that the library contains instances of the two boxes having different spacing?

We now include a table in Figure 1 —figure supplement 1, as well L496-498, which provides further information about the variability of spacer length variants in our datasets. It is worth noting, however, that the best we can do on fully random sequences is show the spacer for the single strongest binding site in each sequence. The whole premise of the thermodynamic modelling used in this work is that many binding configurations on each sequence can matter. So, while the numbers presented in the figure may appear low e.g. for spacer lengths of -2 and +2, states with those spacers always contribute to P_on; the table simply reports cases where that contribution is the strongest. Note that, when fitting the Extended model, we validate that including spacer lengths of -2 and +2 significantly improves predictability.

It will be best to mention clearly the assumptions in the cumulative model, mathematically. A more detailed explanation of the sum in equation 2 will be useful. See for example, in the Bintu et al., paper cited in the same section. Similarly, the MLR model should be described with an equation.

We thank the Reviewer the suggestion. We have now included a set of summary equations into Figure 1, in order to provide a better explanation for what the model is actually doing. We have also provided equations that describe the MLR model (L712-721).

Figure 1 C shows the learned energy matrix. There are other studies which use the standard model to learn their own matrix for RNAP from large mutant libraries. A comparison between the matrices will be useful, even if the promoters are different. In the same vein, this new model can be applied on those data to see if the expression can be better predicted. This will show the generality of the model.

We have included a comparison of our final energy matrix (obtained by fitting the Extended model on all three datasets) and those obtained by Kinney *et al.,* 2010 (Figure 1 —figure supplement 2; for a visual representation see Author response image 1). The elements of the two matrices correlate strongly (r^2^=0.83). Note that the Standard model used in this study is, by construction, the same as that used by Kinney *et al.*, meaning that, if we trained our models on their data, we would observe qualitatively the same improvement of the Extended over the Standard model as we observed in the *P_R_* library (Figure 3).

**Author response image 1. sa2fig1:** 

Does sequence in pR (and pL) contain also the annotated transcription start site (TSS)? Is there any chance that the TSS also gets mutated in their libraries? Since the -10 box is relative to the TSS the distance between the box and TSS is presumably important, which can be defined only with respect to the nucleotide (or a small neighborhood) at the TSS. Does the final energy matrix contain those regions? Figure 1C suggests not.

We thank the Reviewer to point out this complex question. On one hand, the original TSS for the *P_R_* and *P_L_* promoters was not mutated. We clarify this point in the Methods (L456-457). However, we explicitly model for the possibility that RNAP can bind anywhere along the sequence, and some of those binding configurations would have a TSS in the mutated region. We do not model how the changes in the TSS might affect expression levels. However, the fact that the Extended model trained only on the *P_R_* library exhibits high predictability on the random sequences (*36N* library) gives us confidence that the effect of mutagenizing the TSS plays a smaller role in determining gene expression levels compared to the binding energy between RNAP and DNA, at least within our sequences.

I would have expected to see the Y axis of Figure 2B to be stratified across 12 bins. But that is not the case, which means I am missing something.

We thank the Reviewer for pointing out this omission in our explanation of Figure 2B (now Figure 3A). For the *36N* library, because we have 12 bins, we could determine the mean expression (rather than using the median bin as we do in the *P_R_* library). We now clarify this in the figure legend (L1451-1453).

Is it not possible that the energy matrix values (those at the two boxes) change with the spacer? I understand that is not allowed in the current model.

This is a very good point, and we agree with the Reviewer that energy matrix values might change depending on the spacer length. More broadly, any two (or more) parameters in our model could depend on each other. However, understanding such dependencies in a highly multidimensional space is beyond the scope of our work, except for the dinucleotide interactions which we account for. Specifically, for the question raised by the referee, we expect that we would not have enough data to infer energy matrices reliably conditional on individual spacer lengths. We discuss this in more detail in the Methods section (L794-798).

Finally, the contribution of the dinucleotide interactions is not convincing. Perhaps the prediction accuracy on the held out set based on with and without including these terms will help clear that.

In line with Reviewers’ comment 14, we expanded the text and provided a Figure 2D, which includes detailed information about how each extension to the model (including dinucleotide interactions) individually affects predictability on the held out dataset.

Reviewer #3:Lagator et al., extend previous biophysical models to predict gene expression of constitutively expressing *E. coli* promoters from their sequence. They provide evidence that taking into account that RNA-polymerase can bind in multiple configurations, including non-productive ones, significantly improves predictions. They also confirm and extend previous observations that functional promoter sequences are relatively abundant in random sequences.The aim of this study is to develop a realistic biophysical model for predicting the expression of (constitutive) *E. coli* promoters from their sequence. From my understanding of the presentation, the main new results that the authors claim are:1. They extend existing biophysical models to a more realistic model that takes into account that the spacers between the -35 and -10 sites are flexible, that one should not just focus on the best site in a region but take into account cumulative binding to all possible sites, and that one should also explicitly model that binding to some sites is not productive (e.g. binding to reverse-complemented sites on the opposite strand).2. They confirm experimentally that these extensions can indeed have a major impact on expression, for selected promoter sequences.3. This biophysical model accurately predicts the gene expression level of any sequence.4. They show that a substantial fraction of short random sequences will act as promoters, i.e. drive significant expression.5. There is significant evidence for selection against ubiquitous expression, i.e. sigma70 binding sites are depleted both in gene bodies and in intergenic regions.In my opinion the strongest parts of the paper are points 1 and 2. The extensions of the 'standard' biophysical model are all very sensible and the model is implemented in a mostly reasonable way. None of the ingredients in the model are particularly novel or surprising, but that's not a bad thing. I do wonder why the authors did not include features such as allowing only one of the two 'feet' of the sigma70 to bind (like in the Einav and Phillips PNAS 2019 paper) or non-specific binding (although maybe the clearing rate that the authors introduce plays a role analogous to non-specific binding).

The reviewer has a valid point. Extending the configuration space even further to allow bindings of single feet would indeed be possible, and had we started to work on this model after the Einav and Phillips publication, we would have included it in our model. However, by 2019, we have already had our model fitted and all downstream research completed, and had no indication that the effect from including single feet binding would be large.

To investigate the potential contribution of this effect at this stage, we split up the effects coming from each of the feet by calculating the free energy of binding of each foot individually across each sequence in our 36N library, and compared it to the binding energy when both feet bind. The Author response image 2 shows the result of such an analysis, where we see that the FVE in the case of both feet binding (right plot) is much larger than the FVE of either of the single feet binding. We report raw, non-weighted r^2^’s, as well as r^2^’s weighted in the same manner as throughout the manuscript, so that the weights of data points in each bin are inversely proportional to the number of sequences that fall in that bin.

Our analysis of course does not dismiss the reasonable idea the binding of one foot can individually lead to transcription, nor does it address the possibility of avidity between the binding of two feet (as described by Einav and Phillips), but rather suggests it may have a significant – but maybe not very large – influence, yet comparable to other mechanisms that we included. These results are also consistent with the fact that the energy penalties in the -10 box have a higher mean value than those in the -35 box (2.02 vs 1.61), and hence contribute more to the explanatory power.

It also was not entirely clear to what extent binding at one site precludes binding at other sites in the model. This could be explained better.

Our model assumes that RNAP concentrations are not high enough for two molecules to be bound to the promoter simultaneously before they initiate transcription and clear the promoter; under this assumption the Pon can be written as a sum over single-molecule Boltzmann weights. We do, however, explicitly sum over *all* single-RNAP binding configurations on the piece of DNA that constitutes our promoter; that includes binding in configurations that are productive and the ones that are not (e.g., binding on the reverse complement). The logic here is that, on the productive strand, all bound RNAP molecules are released quickly and move in the same direction. In contrast, the released RNAP on the reverse-complement could clash with the productive RNAP. Formally, if the unproductive binding interferes with the probability of RNAP to bind productively via exclusion, the unproductive terms in the partition sum are still linear order in concentration (as opposed to terms where two RNAP are bound simultaneously, which are quadratic and are thus negligible corrections in the regime we consider). Note that the unproductive binding on the reverse complement has the smallest impact on predictability. We provide further clarification on this in the main text (L117-119) and methods (L804-806).

But apart from this the model is very reasonable. The only part I am sceptical about is the dinucleotide modeling, which is based on a very ad hoc and rather baroque procedure.

We agree with the Reviewer’s assessment that the dinucleotide modelling is not carried out using a clean “textbook” inference procedure that we used for other model extensions. This is because of the extremely large number of possible dinucleotide interactions which we did not manage to learn simultaneously in a robust way even with regularization; each pairwise interaction also adds non-trivial computational time and including dozens of interactions already proved computationally limiting. In turn, we devised a more *ad hoc* method that first effectively performs feature selection by finding significant interactions, after which it attempts to estimate their effect. Our main objective when investigating pairwise interactions was to assess how much they could affect predictability and see whether there is any obvious pattern between interactions that we could identify. First, we found that the effect of dinucleotide interactions on the predictability is fairly limited (Figure 2D). Second, there appears to be an interesting pattern where interactions tend to feature basepairs outside of the canonical -10 and -35 feet, as we mention in the main text.

I particularly liked the experimental confirmations (Figure 1 – supplement 1) that support that several extensions of the model can indeed have a major impact on expression for selected promoter sequences. In my opinion these results should probably have been themain focus of the paper, maybe together with some more rigorous quantification of what role the various extensions play in predicting the expression levels of the random sequences. If, in addition, the authors had provided evidence that these extensions are also important for expression in native constitutive *E. coli* promoters, I think this could have been a very nice paper.

We thank the Reviewer for pointing out to us the importance of various extension we included in the model. We now include a new figure (Figure 2) to highlight the experimental confirmations. We quantify their relative impact on predictability in a new table (Figure 2D) and include a brief discussion of this in the main text (L129-137, L226-229).

However, currently the focus seems to be on two aspects that I am much less convinced about. The authors in several places make the claim (point 3 above), that their model accurately predicts expression levels. I do not feel this claim is properly substantiated. First, to give some context, there is a very large body of work going back to the 1980s that aim to predict promoter sequences in *E. coli* directly from sequence, and with just a little bit of googling one will uncover plenty of works in the literature that appear to claim to `solve' this problem, e.g. Cassiano and Silva-Rocha mSystems 2020 present a recent benchmarking of a whole host of promoter prediction tools. I understand that these methods do not specifically aim to predict promoter strength but presumably one could use the 'score' that they calculate as a predictor of promoter strength and it is not clear to me how poorly such predictors would perform in comparison with the model presented here. In addition, there are also several previous papers that specifically claim to successfully predict promoter strength from sequence using either biophysical models close to the ones used here (e.g. Brewster et al., PLoS Comp Biol 2012, Einav and Phillips PNAS 2019) or more general machine learning methods, e.g. Mulligan et al., Nucl Acids Res 1984, Weiler and Wecknagel J Theo Biol 1994, DeMey et al., BMC Biotechnology 2007, Meng et al., Quantitative Biology 2017, and the recent Zhao et al., BioRxiv 2020.06.25.170365. Does the model of the authors significantly improve over these approaches? It is not clear to me. Especially the Einav and Phillips method is curtly dismissed in the discussion, but it is not clear to me whether that model, if trained on this data, would really perform much worse than the current model.

We agree with the Reviewers that we needed to do a better job in demonstrating the predictive power of our model. To this end, we have now analyzed three published datasets – Johns *et al.,* 2018, Urtecho *et al.,* 2019, and Hossain *et al.,* 2020. We selected these datasets because they involved large mutant libraries with 1000s of sequences, and because each of them developed a model to try to predict expression levels directly from sequence. We found that our model led to significant improvements in predictability compared to the models presented by Johns *et al.,* and Hossain *et al.*, while underperforming compared to the machine learning approach used by Urtecho *et al.* It is worth pointing out that this machine learning model is not designed to predict expression from any sequence, as the dataset they based the model on is modular (i.e. they make all possible combinations of pre-set -10, -35, UP element, and spacer sequences). The same applies for the Einav and Phillips 2019 model, which in principle cannot predict expression levels from random sequences because the model is based on the Urtecho *et al.,* dataset. We discuss these findings in the new section of the manuscript (*Testing Model Generalizability* starting at L256).

We would specifically like to highlight that the tests of our model did *not* include refitting the model to different published data sets, but solely an evaluation of the performance of the already fitted model (with the single parameter readjustment for possibly different RNAP concentrations that was inferred for each data set separately). Given our previous attempts (unpublished) with machine-learning-derived models, such across-dataset generalization (as opposed to within-dataset random splits into training and validation data) is a very powerful test that expressive machine-learning models can easily fail even though they show high within-dataset generalization.

We also analyzed the Zhao *et al.,* dataset, but identified some issues with the data. In particular, we found that a large fraction of the reported mutations occur in and around the ribosomal binding sites. Our model does not account for the effects of mutations in ribosomal binding sites, resulting in low predictability. For this reason, we decided not to include this analysis in the manuscript, but provide Author response image 3 for your and Reviewers’ reference.

**Author response image 3. sa2fig3:** 

We did not analyze any of the other datasets mentioned by the Reviewer above, because all of them contained only a small number of promoters (not more than 100), preventing the authors from developing persuasive models for predicting gene expression levels from sequence.Finally, we did not develop a predictive method based on bioinformatics approaches that aim to identify promoters (as opposed to predict gene expression levels), because these models (as outlined by Cassiano and Silva-Rocha 2020) assume that a promoter has a single dominant RNAP binding site, as well as not accounting for the energy impact of spacer length variability. While constructing a model along these lines would be possible, it would be very labor intensive and would most likely look very similar to the Standard model we used.

More broadly, we would like to point out – and that we now emphasize much more clearly throughout the manuscript (for example, L89-99, L157-170) – that our study implicitly already compares the performance of the Extended model to various other studies (such as Brewster *et al.,* 2012 and Kinney *et al.,* 2010), through the comparison with the Standard model. The Standard model works under the same assumptions used by all existing mechanistic models aimed at predicting gene expression levels from any sequence (like Brewster *et al.,* 2012 and Kinney *et al.,* 2010). Similarly, it works under the same assumptions as most bioinformatics models for predicting promoters, as they assume that a single RNAP binding site is what defines a promoter. Hence, the comparison between the Extended and the Standard model implicitly already informs about the Extended model performing better than the existing mechanistic approaches. Moreover, the Standard model is given the “best chance” since (i) we fully optimize it on our datasets; (ii) allow it to use a flexible spacer (but with no energy penalty for different spacer lengths); (iii) allow it to use a larger binding footprint than the standard 6+6 basepairs or the matrix by Kinney *et al.* In this sense, any excess performance by the Extended model over Standard model that we report is a conservative estimate for what could be expected if we were to use a previously published mechanistic models with standard features.

This reasoning, of course, is true only for a comparison with mechanistic approaches. While some machine learning approaches might outperform our model on the specific datasets on which they were trained on, we are not aware of such models that work for any random sequence and generalize well across datasets; in particular, irrespective of their predictive power, such models would lack the mechanistic interpretability that we desired.

In response to this, and other, Reviewer comments, we have softened our language and removed the use of the subjective term ‘accurate’.

I understand it is kind of a pain to do extensive comparison with previous methods but in order to make real progress on understanding what determines the expression levels of constitutive *E. coli* promoters, then at a minimum I would have expected a very clear quantification of exactly how well the model performs. What one would want to know is: If I feed the model a random sequence, how accurately does it predict its expression level? Or, how well can the model predict the variation in expression levels of constitutive promoters across the *E. coli* genome? How much accuracy will be lost when various of the extensions are removed, either on native promoters, or on random sequences.

We agree with the Reviewer that we should have been clearer and done a more thorough job in validating our model. The Reviewer raises three ways in which we can provide more details and validations of Extended model performance:

a) “If I feed the model a random sequence, how accurately does it predict expression levels” – we explicitly test this by validating the Extended model on the validation portion of the *36N* library (and now clarify in various locations throughout the manuscript that this is explicitly what we have done).

b) “How well can the model predict the variation in expression levels of constitutive promoters across the *E. coli* genome” – to this end, we now include a comparison of the Extended model performance on three previously published datasets, and find that the model performs well on all of them (see section *Testing Model Generalizability)*. It is worth noting that this performance was achieve without any fitting of the Extended model to these new datasets (with the exception of chemical potential). We did not examine the performance of our model on constitutive *E. coli* promoters because the existing datasets measuring expression levels of constitutive promoters come from RNA-seq experiments, which are notoriously imprecise.

c)“How much accuracy will be lost when various of the extensions are removed” – we provide more detail on how each model extension contributes to overall predictability (Figure 3D, L129-137, L217-229).

In order not to overstate our findings, we toned down the language, especially by minimizing the use of the word ‘accurate predictions’ and instead only talking about ‘predictions’.

However, we do not really get such an analysis. The authors only present one scatter and Pearson correlations between the predicted log[Pon] and measured log[expression] for various versions of the model. Crucially, I do not think that a correlation of r^2^ ~ 0.75 on a dataset where expression levels vary over a 1000-fold range can be reasonably described as `accurate prediction’. Just as a back-of-the-envelope, if both data and model have a standard-deviation in log-expression levels of about log[100], then r^2^ = 0.75 corresponds to the error between model and data having standard-deviation of log[10]. That is, unless I made a mistake in my back-of-the-envelope, the model’s predictions would typically be off by 10-fold (and the scatter does seem sort of consistent with this). T is also in line, with the fact that the bins for the P_R data are 10-fold apart in expression level, i.e. the training of the model is done on a dataset in which promoters whose expression levels are within 10-fold of each other are effective treated as equally expressed. I do not think that this level of precision can be called accurate prediction of expression levels. For example, using very similar reporter constructs, the difference in expression of the median and 95^th^ percentile of native *E. coli* promoters is about 20-fold (i.e. see Wolf et al., Figure 1B), so this model could barely tell these apart.

The reviewer correctly notices that the *P_R_* (and *P_L_*) datasets consist of wide bins, which encompass sequences potentially 10-fold apart in expression level. Indeed, being able to correctly predict bin identity of a sequence with a 10-fold error should not have been particularly surprising. However – and in part because of that – we also apply the same biophysical model to an independently obtained and more precise dataset (*36N*), where we report higher performance, suggesting that the performance on *P_R_* / *P_L_* datasets is limited by the measurements themselves. Moreover, even a model fitted only on the *P_R_* dataset performs well on the dense and independent *36N* library, indicating that the error of the model predictions themselves cannot be as high as 10-fold. Below, we make the argument more precisely for the *36N* dataset.

Let V be the variance of the measured log10(expression), and MSE the mean square error of the prediction of the same quantity. The fraction of variance explained (r^2^ in the case of a linear fit) is then FVE = 1-MSE/V, so a single theory is evaluated differently depending on the variance of observations V. Our ideal dataset is uniformly sampled across all possible expressions (in particular, the distribution over expressions is not Gaussian), and to approximate that, we introduce weights so that the total weight in each log-equidistant expression bin is equal. For our random library (*36N*) weighted in this way, the variance of log10(expression) V = 1.016. A value of FVE=0.75 yields the MSE of ~0.25, i.e. the error of ~0.5, which corresponds to a 3-fold error in expression. This might not be outstanding, but one should think of it in the context of the 1000-fold dynamic range of measurable expressions that we probe.

For reference, a uniform distribution of log10(expression) spanning a 1000-fold range has V=0.75; FVE of 0.75 in this case would yield an error of 0.43 for log10(expression), i.e. 2.7-fold for expression; this is close to the calculation for our data, above. The reviewer does the same back-of-the-envelope calculation, assuming standard deviation log10(100), i.e. V=4, thus arriving to a 10-fold error. We agree such a precision would be much less remarkable than what we can report.

At this point we would like to thank the reviewer for challenging us to think of performance in terms of the errors, rather than r^2^. Consequently, we have included this discussion in the main manuscript (L232-234) in order to explicitly clarify to readers what our r^2^ = 0.8 means for individual errors and model accuracy. We make it explicit that it corresponds to a 3-fold error in a dataset spanning expressions from very strong promoters, down to not detectable. Among reported promoters in *E. coli*, which naturally have a narrower range of expression, the same error yields a much lower FVE.

Another issue is that it is not clear which extensions of the model are the most crucial for the accuracy of the predictions. I was surprised to see (in Figure 2 – source data 1) that, apparently, by far the biggest improvement on the random sequences (36N dataset) comes from including flexible spacers and that cumulative binding and occlusive binding adds rather little. This appears somewhat at odds with the narrative that the authors present in the paper that stresses the importance of cumulative binding.

We expanded the discussion of the relative contribution of each extension (L129-137, L217-223), added a new figure to main text (Figure 2, especially panel D) and have also altered the language throughout the manuscript to not insinuate that cumulative binding is more important than other extensions, especially when compared to spacer flexibility. It is worth pointing out that cumulative binding enables multiple spacer length configurations to co-exist within the model, compared to the Standard model which allows only for a single configuration. Hence, the two are intrinsically linked. However, the Reviewer is right to point out that when evaluated individually, spacer flexibility contributes more to overall predictability of our datasets. We see cumulative binding as a necessary basis on which other effects may or may not build upon.

Importantly, cumulative binding plays a bigger role in determining lower expression levels (see Figure 4 —figure supplement 2). This means that the total r^2^ contribution across all sequences could be small while the impact of cumulative binding on weakly expressing sequences is large. Furthermore, cumulative binding has an effect of shifting the mean expression of a random sequence to higher values, with clear evolutionary consequences that we explored in Figure 4. Specifically, the contribution of cumulative binding to low expression levels plays a key role in promoter evolution by providing a greater likelihood of a random mutation leading to an increase in expression, especially when evolving from non-functional sequences (L347-351).

In short, although I would like to believe that this reasonable biophysical model improves a lot over previous approaches in the literature, I don't think a convincing case has been made that this is really a major advance in accuracy of promoter strength prediction.

We hope that the additional analyses included in the revised manuscript persuade the Reviewer that our model either performs better than the existing approaches, provides more mechanistic insights into the functioning of promoters, or both. We have expanded the explanation of the model to include the arguments presented by the Reviewer throughout our revised manuscript.

Finally, regarding points 4 and 5, I do not understand why the authors place so much emphasis on the observation that functional promoters (i.e. sequences that drive gene expression) are common among random sequences, because this is neither novel nor surprising. It has been long-known that the 'position specific weight matrices' that model sigma70 binding sites have low information content, e.g. Schultzaberger et al., Nucl Acids Res 2007 estimate it to be about 6.5 bits. Given this, one would very roughly expect a site every 100 bp in random sequences. In fact, this was one of the reasons why, in Wolf et al., (eLife 2015), we were confident that we would easily find a large diversity of functional promoters in a library of random sequences. This expectation was confirmed by our measurements as one can see from Figure 1B of Wolf et al. That is, after selecting the 5% of highest expressing cells from the random library, most of them express at a reasonable level. At the time it didn’t even occur to us to mention this as a surprising observation. When the paper of Yona et al., (Nat Comm 2018) appeared, we realized that we misjudged this, and that for many people it WAS surprising that functional promoters are common even in relatively short random sequences. However, now that Yona et al. have an entire paper focused on this observation, including showing depletion of sigma70 sites within gene regions, there is no more novelty to this observation.

We appreciate the concerns that the Reviewer has regarding the novelty of the ‘evolutionary’ set of results presented in our manuscript. It is clear that in the original submission we failed to appropriately emphasize the evolutionary relevance of our results.

First, and most importantly, we specifically compare the predictions about promoter evolution derived from the Extended model to the predictions from the Standard model. This is precisely because the Standard model already captures exactly the assumption that the Reviewer and Yona *et al.,* are talking about – it reflects the standard, textbook view of how bacterial promoters work. Namely, the prediction that you would get roughly 1 RNAP binding site in every 100bp is based on the standard position weight matrices of σ^70^-RNAP and the expression expectation is based on a straightforward view of how binding maps into expression. The Extended model, which captures various additional features of promoter architecture, predicts many more expressive binding configurations *compared to* the Standard model, which shares the assumptions discussed by the Reviewer. We are thus not making a verbal claim about whether there are “a lot” or “a few” expressing sequences among random sequences *per se* (such a qualitative statement is anyway somewhat arbitrary in what is considered “a lot”); rather, we are making a quantitative and comparative statement that the Extended model predicts much more expression from random sequence relative to the Standard model – even if the latter already predicts “a lot”. RNAP binding is more promiscuous than predicted by the Standard model. For quantitative details on how much faster promoter evolution is when predicted with the Extended model, see Figure 4D.

Second, evolutionary simulations in Figure.4D show further that such differences between the Standard and Extended model can be drastically potentiated by the evolutionary dynamics. Even though we only expect ~3-fold change between Standard and Extended model in the fraction of sequences that measurably express (Figure 4A), this can map into multiple-order-of-magnitude changes in the time-to-evolve under the SSWM evolutionary model. This is because of the multitude of expressing states and evolutionary paths that are available to the Extended model but forbidden to the Standard model. This is most clearly seen when expression has to evolve under constraints (Figure 4D, when only a small region is mutated). This point is not trivial at all: in Tugrul *et al.,* 2015 we showed that the predicted dynamics of evolution of “standard” TF binding sites (with no spacer flexibility and other features of the Extended model) is worryingly slow and difficult to square with the reported estimates for the timescales of regulatory evolution based on comparative genomics studies. There, the authors failed to identify biophysical mechanisms that would enable faster evolution of regulatory sequence. Here, in contrast, starting with data, we see that the particular properties of RNAP binding captured by the Extended model do result in much faster promoter evolution (compared to the Standard model expectation). Therefore, we believe that these results provide a much wider relevance to our work than simply reporting improved predictability due to the inclusion of a host of biophysical mechanisms into the model.

We have now changed the text in multiple places to reflect and emphasize these points (most substantively in L342-351).

It is also worth pointing out that we provide a deeper basis for the ‘evolutionary’ findings that we report compared to Yona *et al.,* while also exploring several novel aspects of promoter evolution. For example, we show not only that σ^70^-RNAP sites are selected against in the coding regions (which was shown by Yona *et al.,* as well) but also within *promoter* regions themselves, even when we only use the list of verified promoter sequences from RegulonDB. To us, this seems like a novel finding, that corroborates the promiscuity of RNAP binding under the Extended model. Similarly, it seems relevant to report findings about promoter evolution even if they show the same results as Yona *et al.,* did, because we base our findings on a vastly larger dataset combined with a model capable of actually explaining much of the diversity we observe in that data, as opposed to being based on a sample size of ~40 mutants.

For all of these reasons, we believe there is substantial novelty in our results. We have changed the manuscript in a variety of places to better emphasize what is novel about our results and to properly explain that the Standard model actually represents the traditional view of promoters (most substantively in L89-99).

The observations about the frequency of occurrence of promoters in random sequences could have been more interesting if it had been more meaningfully quantified. That is, any sequence will presumably drive expression at some nonzero rate. An interesting (and meaningful) quantitative question is how likely it is that a random sequence drives expression at rate > x, as a function of x. Maybe the authors' data and model could give a reasonable answer to this question.

This is a great suggestion and we have now included an additional plot in Figure 4A, which shows the cumulative distribution function (CDF) for the Extended model predictions of expression from random sequences (as well as the CDF for the experimental flow cytometry measurement of the *36N* library). This plot shows the likelihood of finding a random sequence with measurable expression above any set threshold.

However, instead the authors simply treat this problem as if there is just a binary question whether a promoter is expressing or non-expressing. Moreover, they define the threshold between expressing and non-expressing in a way that is not biologically meaningful but seems determined by instrument precision. I actually had a hard time understanding how precisely 'measurable expression' was defined. My understanding is that, for the P_R and P_L libraries, it is defined as any expression higher than the 99th percentile of measurements for cells carrying no YFP. But how it is defined for the 36N library (with another reporter) was unclear to me.

We apologize that our explanation of how we defined ‘measurable expression’ wasn’t clear. The threshold is defined the same way for all three libraries – as the 99^th^ percentile of the distribution of fluorescence measurements from cells carrying no fluorescence marker of any sort. The only difference is that in the *36N* library, we include one additional cell sorting gate below the detectability threshold. In modelling, we had to assume that the thermodynamic assumptions behind the model apply the same way to sequences below measurable threshold. We have now clarified these points, both as they relate to experiments and model, in the legends to Figures 3 and 4, as well as the *Sort-seq experiments* section of the Methods (L520-523, L526-528).

We do appreciate the reviewer’s comment that the simplistic use of the language of whether there is a promoter or not (as opposed to a quantitative distinctions in its strength) is inappropriate, and we have carefully reworded a number of sentences to clarify.

In any case, it appears that 'measurable expression' is defined in terms of the auto-fluorescence level that the cells have in the FACS machine. Unfortunately, this is not a biologically meaningful cut-off. In fact, a substantial fraction of *E. coli* promoters express below the detection limit in the FACS (i.e. not significantly above auto-fluorescence).

We agree with the Reviewer, and have now added a caveat explaining this point in the main text. Specifically, we explain now in the main text that we assume that the energetics of σ^70^-RNAP binding to DNA drive promoter activity below the instrument detectability threshold as they do above it (Figure 3 legend – L1456-1458).

In summary, I think there are some very good ideas in this paper. I in principle like the biophysical model, and the experimental validations of the model's extensions I thought were very nice and, when extended, could make a compelling paper. Unfortunately, the current focus of the papers appears to be on the frequent occurrence of promoters in random sequences (which is not novel) and claims that the model is a major advance in predicting gene expression from sequence, which I do not feel are substantiated by the results presented.1. I had real trouble understanding the processing of the 36N dataset. I understand that, because in the 12-bin dataset you sort until you have a certain number of cells in the bin, you need to correct for this when estimating the relative fractions of cells with a given promoter that go to different bins. But I do not understand the approach with the spike-ins. Aren't there, in addition to the spike ins, also an unknown number of reference promoters coming from the sorting?

We understand the Reviewer’s confusion about the spike ins, as we hadn’t explained this procedure well enough in the original submission. Now we clarify in more detail in the Methods (L645-648) how we introduced the spike ins – the spike ins were done using a reference Λ *P_R_* promoter, while the plasmid used for cloning the *36N* library (wildtype plasmid) did not carry the Λ *P_R_* promoter. Hence, we could easily distinguish between the reference plasmid we added, and the wildtype plasmid that was not cloned properly and was hence not carrying promoter mutations.

2. line 81: "even for constitutive promoters, where σ70-RNAP binding solely determines gene expression levels" Has this really been established? There could be effects of differential binding of topoisomerases that undo supercoiling.. or differential weak binding by other TFs.. and so on. I don't think we really know these levels are solely determined by sigma70-RNAP.

We now discuss in the main text the caveat identified by the Reviewer (L80).

3. Figure 1 – supplement 1.Why is detectability threshold higher in B than in A? I found it generally quite frustrating that it is hard to figure out how exactly this 'measurable expression' threshold, which plays such a crucial role, is defined.

We thank the Reviewer for spotting this error. There was a formatting error with the figure which we have now fixed. We have also included an explanation for how ‘measurable expression’ was defined for this specific set of measurements in the legend to Figure 2 (which used to be Figure 1 —figure supplement 1), as it is defined differently for populations measured in a flow cytometer and those measured in a plate reader. Having said that, the logic behind the definition is the same – measurable expression being defined by the fluorescence level of the strain carrying no fluorescence markers.

4. 515-516: Finally, we required the distribution of expression bins to be as unambiguous as possible (Figure 2 —figure supplement 1D,E).Why is this done? The way I understand it, this throws out promoters whose expression levels straddle two bins. I guess you want to throw those out because the fitting of the model to only observations in 4 bins only is more accurate when using only promoters that fall sort of in the middle of those bins, but I find this quite unsatisfactory. Would it not have been possible to fit a log-normal to the relative fractions of counts in the different bins and get continuous expression estimates?

By requiring unambiguity, we only meant that we discard sequences with standard deviations larger than 0.5, following with the requirement that the mean, median, and mode of the distribution across bins should point to the same bin. For the P_R_ library, the first step reduces the number of sequences from 29020 to 22884, and the second one further to 22769. The same numbers for the P_L_ library are 6415, 4239 and 4222. We’ve amended the text to make this fact more explicit (L699-706).

As for the possibility of using the relative fractions, we opted not to follow that route because of the abundant possibilities for introducing biases during library preparation. From the moment the cells are sorted into their appropriate vials to when we obtain their sequences, they undergo several passages (overnight growth and subsequent dilution) to allow populations to grow large enough numbers so that sequencing is more robust. However, such passaging can introduce a bias in the form of over- or under-representation of a mutant in one bin compared to other surrounding bins. PCR adds further potential for departure from the assumption that the two reads are independent and identically distributed. An additional benefit was that it simplifies the analysis and interpretation. This explanation has now been added to Materials and methods (L596-602).

5. 596: "We define 60:20:20% splits of each of our libraries into three disjunct datasets Figure 2 – Source Data 3."Are these splits random? A notorious problem in cross-validation based fitting is when highly correlated datapoints occur in both training and test data. For example, in this case you could have promoters that are virtually identical, and with equal expression, in both training and test datasets. This could lead to misleadingly high performance on the test dataset. I actually doubt this is happening here, but maybe it is worthwhile checking this.

Due to the nature in which we process our libraries, we are dealing with unique sequences. Namely, we pick each random sequence that appears in the entire sequencing output only once, and use the median bin to represent its fluorescence (in the *P_R_* and *P_L_* libraries) or the actual mean bin (*36N* library). This sequence then gets assigned either to the testing or training data, but not both; hence, in our dataset, the splits are disjunct by design.

Sequences in the testing and training sets are correlated (but not identical) in the *P_R_* and *P_L_* libraries by construction, since all sequences are derived from the same wildtype; sequences are uncorrelated in the 36N library. The fact that our model performs well when tested also on different libraries from the *P_L_* on which it was fit should provide strong evidence against any effects that the referee is referring to.

We clarify this point in more detail in the Methods section *data splits and fitting procedure* (L689-691).

6. Did you try adding a non-specific binding term to the model? Or the ability for sigma70 to bind through only one of its two feet? If not, why not? Do the authors think sigma70 cannot be bound only at the -10 or -35 sites?

To properly address this concern, we need to consider two different scenarios. First, that the binding of σ^70^ through only, for example, -10 foot, does not alter in any way its binding preference. In other words, the energy matrix for the -10 foot is the same no matter whether σ^70^ is bound only with the -10 foot or with both, -10 and -35 feet. Our model allows for this scenario fully. In fact, our model does not ‘know’ whether the binding occurs predominantly through one foot or the other. All it cares about is the total binding energy, which one can achieve through ‘mix and match’ of positions throughout the entire energy matrix. If, however, this assumption is not true, then the sequence preference of, for example, the -10 foot, would depend on whether and to what extent the -35 foot is bound. To explore for this possibility, we would need to re-fit all model parameters under at least two additional binding configurations – -10 foot bound alone and -35 foot bound alone, and this kind of analysis was beyond the scope of our study. Just to illustrate how much more difficult and time-intensive this task would be, consider that the Einav and Phillips 2020 PNAS paper was entirely dedicated to fitting only this one aspect of polymerase binding, and it did so with only eight -10 and eight -35 variants. We have thousands of sequences variants, which would make this work substantially more complicated. However, it would be very interesting to pursue this line down the road.

7. I did not like the way multinomial logistic regression was used to fit the model. Expression should go up monotonically with Pon. It makes no sense to treat the bins as unrelated 'classes'.

This comment is very similar to that of the Reviewer 1, so here we repeat our response: We implemented one-vs-rest MLR as a conservative choice, which would allow for a non-monotonic relationship between P_on and expression. Ordinal logistic model would force it by definition. In particular, for all the models without the clearance rate, if there were a binding state so strong that the expression is actually lower than for a bit weaker binding, we would be able to detect it (and we were actively looking if any such effect exists; it does not). Despite using MLR, we find a monotonic relationship between P_on and expression, which we took to be one of our internal consistency checks.

8. Comparing evolution of promoters de novo under the extended model against the 'standard model' is a straw man in my opinion. The standard model is an approximation to the thermodynamic model that is valid when there is one site that dominates the overall affinity of a promoter. I agree that it is often assumed that this is a good approximation for native promoters that have been shaped by selection. But I do not think there is anybody that thinks this is a good model for how promoters appear in random sequences. Obviously weak promoters can then appear anywhere in the sequence.

We are glad the Reviewer pointed this lack of clarity in our explanation of how we actually use the Standard model. We do not assume that it has to bind to a specific location in the sequence. In fact, we scan the entire sequence for all possible binding configurations, just like we do with the Extended model, but with the Standard one we then only pick the single strongest binding configuration (as opposed to summing up across all binding configurations like with the Extended model). As such, when simulating promoter evolution, we do not require the Standard model to have a fixed position. Rather, we require it only to have only one binding site, no matter where that site might appear in the sequence. We now clarify this point further in the Legend to Figure 4, as well as the main text (L242-245).

9. For Figure 3A, I think it is essential to also predict such a curve for the sequences in each of the libraries and see how well these predicted curves match the observed distributions for these libraries. Otherwise it is hard to judge how meaningful this predicted distribution is.

We have now predicted the distribution for random sequences using the Extended model and compare it to the experimentally measured distribution of the *36N* library, as shown in Figure 4A.

10. Many intergenic sequences in *E. coli* will be constrained to have all kinds of TFBSs, precluding sigma70 to occur in those positions. So to check whether there is really depletion of sigma70 sites, comparing with random sequence of the same nucleotide composition is not quite correct in my opinion. First, you should distinguish intergenic regions that are upstream of two operons from those that are upstream of one, and those downstream of two promoters. Those 3 classes of intergenic regions are also known to have different nucleotide composition. The best regions for tests would be downstream regions, and regions that are upstream of an operon that is constitutive.

We agree with the Reviewer that the way in which we defined promoters in Figure 4E is neither ideal nor comprehensive. What we wanted to do is check for our hypothesis on the two ends of the spectrum – a relaxed definition of what a promoter is (which is what we used) and a very conservative definition (only experimentally confirmed promoters). Our relaxed definition includes all three ‘classes’ of promoters, including those that are found downstream of operons and hence are unlikely to act as promoters. In order to eliminate all intergenic regions that do not contain a promoter (including all the ‘downstream’ promoters mentioned by the Reviewer), we use the conservative definition that includes only the intergenic regions with known promoter function taken from RegulonDB. We use this conservative definition because we are asking whether there is selection against σ^70^-RNAp binding sites, without discriminating the source of that selection, which can be the existence of one or more transcription factor binding sites and/or selection against having too many σ^70^-RNAp binding sites.

To supplement our approach, the Reviewer suggested to do the following: to split intergenic regions into three categories and to look for evidence of selection against σ^70^-RNAp binding sites in two of those categories. First is the promoters downstream of operons. These are intergenic regions expected not to have any promoter activity. It is unclear to us how providing evidence of selection against σ^70^-RNAp in such regions would provide novel insights compared to what we already identify by examining selection in coding regions. Furthermore, we already include these regions in our relaxed definition of a promoter in which we demonstrate strong evidence of selection against σ^70^-RNAp binding sites.

The second set of promoters that the Reviewer suggests to look at, are the upstream regions of a constitutive operon. This definition poses a few technical difficulties for our analysis. First, it is difficult to define a constitutive promoter. Definitions based on experimental evidence are inevitably at the mercy of the specific media conditions used, which in turn define the transcriptome and hence define which promoters might be defined as constitutive. Depending on how many and which environments are tested would therefore give quite a different set of promoters deemed to be constitutive. As carrying out a comprehensive experiment in all possible environments is, obviously, not possible, the best we can do is to look at all promoters reported in RegulonDB or other similar aggregate databases that accumulate information from multiple published sources. ~80% of all σ^70^
*E. coli* promoters reported in RegulonDB have at least one associated transcription factor, meaning that following what the Reviewer suggested would result in us analyzing only ~20% of the sequences included in our conservative promoter list. This number of promoters is very low, meaning that we would not have enough statistical power to discern any but the strongest of selections against σ^70^-RNAp. Importantly, we set out to analyze the evidence for selection against σ^70^-RNAp sites, irrespective of the source of that selection, so we consider it important to consider the promoters that bind transcription factors rather than focusing only on constitutive ones.

We hope that the fact that we found the same evidence for selection against binding sites even within only the known, experimentally confirmed promoters obtained from RegulonDB (Figure 4 —figure supplement 3B) reassures the Reviewer that our findings do hold no matter how we define a promoter. We have added a discussion about the conclusions that can be drawn from our approach with respect to the source of the selection we report (L368-372).

Reviewer #4:The manuscript by Lagator et al., examines a general framework to predict the constitutive level of gene expression for bacterial DNA, which constitutes one of the most fundamental processes in biology. Whereas previous work has typically been restricted to a limited sampling around a well-studied promoter (such as the lac operon), the authors push towards a model that can characterize all possible sequences, which they support with libraries containing 10,000 mutants each around two strong promoters (PR and PL) as well as a library containing 10,000 random sequences (36N). This work represents an important advance to the field, and will hopefully serve as the stepping stone for better models that include transcription factors and eukaryotic enhancers.

We thank the Reviewer for the encouraging comments.

While the authors provide a clear and compelling story using their impressive data set, the one piece that I felt was missing was a connection with large gene expression data sets from other groups. How robust will their already-trained model be when applied in another context, where the experimental design is somewhat different? This will help other groups understand the potential gain in implementing this model for their own ends, and if any fine-tuning or model adjustments need to be performed, such information would be important to add. To this end, I suggest a few possible data sets below. The authors do not need to analyze all of them (although more is better), and they are welcome to substitute other data sets of their choice:1) Hossain 2020 [https://doi.org/10.1038/s41587-020-0584-2] analyzed 4,350 bacterial promoters with a broad range of gene expression measurements varying by 820,000-fold. These promoters should be constitutive, although their random DNA sequences might occasionally give rise to transcription factor binding sites (as in the 36N library). The sequences and read counts are available in their Supplementary Data 3.2) Urtecho 2019 [https://doi.org/10.1021/acs.biochem.7b01069] analyzed the constitutive expression from 10,000 promoters. Their setup ensures that no transcription factors are involved, providing a clean test for the model. You can find their expression data at the GEO Accession Number GSE108535 [https://www.ncbi.nlm.nih.gov/geo/query/acc.cgi?acc=GSE108535]. The file "GSE108535_sigma70_variant_data.txt.gz" contains the names of promoter variants and the average RNA/DNA expression levels of two biological replicates, while the file "GSE108535_barcode_mapping.txt.gz" contains the name of each promoter variant and its corresponding sequence (where the sequence is specified in the column 'most_common'). Figure S12 and Table S1 from their supplementary information provide a visual schematic for how to put the pieces together.3) Johns 2018 [http://doi.org/10.1038/nmeth.4633] analyzed 30,000 promoters mined from prokaryotic genomes, which would be a very interesting avenue to explore. These promoters may involve transcription factors (although their mining strategy might bias them towards constitutive promoters). Their sequences are given in Supplementary Table 1 and their expression measurements for *E. coli* are provided in Supplementary Table 3.

As suggested by the Reviewer, we have now analyzed the three datasets and evaluated the performance of our model. We did so without any additional fitting (except for the chemical potential, which depends on the strain, growth conditions and other factors influencing the concentration of RNAp inside the cells), and we were very encouraged by the high performance of our model across all three datasets, especially as they were all developed to address very different questions. These datasets were also especially useful as each had a model developed by the respective authors, allowing us to compare our model to their performance. The manuscript now contains a new section discussing these results (*Testing Model Generalizability*).

I suspect the authors are already planning to do this, but it would be very helpful if a program implementing the fully-trained model was provided (along with some examples of its usage), to help the community utilize these results.

The code related to this manuscript has been deposited on github and can be found at https://github.com/szarma/Thermoters.